# The Role of Selected Chemokines and Their Receptors in the Development of Gliomas

**DOI:** 10.3390/ijms21103704

**Published:** 2020-05-24

**Authors:** Magdalena Groblewska, Ala Litman-Zawadzka, Barbara Mroczko

**Affiliations:** 1Department of Biochemical Diagnostics, University Hospital in Białystok, 15-269 Białystok, Poland; magdalena.groblewska@umb.edu.pl; 2Department of Neurodegeneration Diagnostics, Medical University of Białystok, 15-269 Białystok, Poland; ala.litman-zawadzka@umb.edu.pl

**Keywords:** glioma, central nervous system tumor, chemokine, conventional chemokine receptor, atypical chemokine receptor, angiogenesis, inflammation, leukocyte

## Abstract

Among heterogeneous primary tumors of the central nervous system (CNS), gliomas are the most frequent type, with glioblastoma multiforme (GBM) characterized with the worst prognosis. In their development, certain chemokine/receptor axes play important roles and promote proliferation, survival, metastasis, and neoangiogenesis. However, little is known about the significance of atypical receptors for chemokines (ACKRs) in these tumors. The objective of the study was to present the role of chemokines and their conventional and atypical receptors in CNS tumors. Therefore, we performed a thorough search for literature concerning our investigation via the PubMed database. We describe biological functions of chemokines/chemokine receptors from various groups and their significance in carcinogenesis, cancer-related inflammation, neo-angiogenesis, tumor growth, and metastasis. Furthermore, we discuss the role of chemokines in glioma development, with particular regard to their function in the transition from low-grade to high-grade tumors and angiogenic switch. We also depict various chemokine/receptor axes, such as CXCL8-CXCR1/2, CXCL12-CXCR4, CXCL16-CXCR6, CX3CL1-CX3CR1, CCL2-CCR2, and CCL5-CCR5 of special importance in gliomas, as well as atypical chemokine receptors ACKR1-4, CCRL2, and PITPMN3. Additionally, the diagnostic significance and usefulness of the measurement of some chemokines and their receptors in the blood and cerebrospinal fluid (CSF) of glioma patients is also presented.

## 1. Introduction

Chemokines are small structurally related heparin-binding proteins whose name comes from the term “**chemo**attractant cyto**kines**” and refers to their originally described function as the stimulators of the migration of various types of cells [1]. These proteins are also synthesized by various types of cells, including monocytes, macrophages, T lymphocytes, neutrophils, and fibroblasts, as well as neural, endothelial, and epithelial cells [2].

Chemokines and their receptors play important roles in the development of various tumors by promoting proliferation, survival, metastasis, and neoangiogenesis. Although the role of some chemokine/receptor axes in pathophysiology of cancer is well recognized, little is known about the significance of atypical receptors for chemokines (ACKRs) in brain tumors, especially in gliomas. The objective of the study was to present the role of chemokines and their conventional and atypical receptors in CNS tumors. Therefore, we performed a thorough search for literature concerning our investigation via the MEDLINE/PubMed database.

## 2. Materials and Methods

### Literature Search and Data Extraction

We performed a comprehensive search of the literature using the PubMed electronic database from November 2019 to April 2020, with the key words “cancer AND chemokine AND receptor” (8566 papers). Then we narrowed our search to following key word sets: “glioma AND chemokine” (720 items); “glioma AND chemokine receptors” (343 results) as well as “glioma AND atypical chemokine receptors” (55 studies) and “angiogenic switch AND chemokine” which gave 33 papers. A total 1151 records were found. Then we excluded 785 articles not containing data useful for our analysis. In a result, we obtained 366 remaining full-text articles that were assessed for their eligibility. In the final step, we excluded 112 review papers, and so 254 original publications were included in the study (Figure 1—PRISMA 2009 flow diagram [3]).

## 3. Chemokines and Their Receptors

Chemokines are small, mainly soluble, secretory proteins, which upon secretion are immobilized on endothelial cells and in proteoglycans and glycosaminoglycans (GAGs) within extracellular matrix (ECM) [4]. This immobilization facilitates the generation of a specific concentration gradient, which is crucial for chemoattraction and migration of target cells in a directed way. Embedded in the cell surface CXCL16 and fractalkine (CX_3_CL1) are the exception in this regard but they also can be cleaved by metalloproteinases, what results in the release of their active parts into the extracellular space [5].

The complex chemokine network consists of over 50 ligands, which share a conserved protein structure within N-terminal domain with two cysteine residues, designated as “chemokine scaffold” [6]. Chemokines may be divided into four subfamilies based on the positioning of these residues: CXC subfamily with cysteine residues separated by a single AA; CC chemokines, with two adjacent cysteine residues; XC chemokines, which have only a single cysteine residue in the amino-terminus; and CX_3_C subfamily with three AA residues separating the cysteine tandem [7]. The regulation of chemokines activity is based on the network of feedback loops and certain mechanisms responsible for their suppression and/or stimulation. They also control the range of chemokine levels in the extracellular fluid.

Binding of chemokine to membrane receptors triggers activation of various intracellular signaling pathways. All 25 members of chemokine receptors (CKRs) belong to the G-protein-coupled receptor (GPCR) family. Chemokines receptors also share a conservative structure with 50% homology within the same class and 30% between different classes [8]. They include seven transmembrane domains connected by extra- and intracellular loops. All CKRs have also two conserved cysteine residues linked by a disulfide bridge. Binding chemokine to the receptor initiates the dissociation of G-protein subunits, followed by the activation of mitogen-activated protein kinase (MAPK), phosphoinositide 3-kinase (PI3K), and phospholipase C (PLC) pathways and the increase in the intracellular calcium levels [9,10].

Conventional chemokine receptors (cCKRs) are named according to the predominant type of chemokine recognized and may be divided into three subclasses: there are six receptors binding CXC chemokines (CXCR1–6), 10 receptors recognizing CC chemokines (CCR1–10), and two receptors for XC chemokines (XCR1 and CX_3_CR1) [1,11]. Chemokine interactions with their receptors are characterized with some promiscuity: chemokines in one class can bind to several receptors from the same class [2], while CKRs also can have multiple ligands within given class of chemokine [12]. There are only a few ligand-receptor pairs that are clearly defined as highly specific, such as CCL20 and receptor CCR6 or CXCL13 with receptor CXCR5 [13].

## 4. Biological Functions of Chemokines

Apart from chemotaxis, a capacity to induce a directional migration of leukocyte subclasses towards a chemokine gradient, which was the first recognized and described function of chemokines [14], these proteins may play many other physiological roles. Although they have pleiotropic properties, chemokines may be divided functionally into two main groups: inflammatory or homeostatic chemokines. Some chemokines can be classified to both types, based on their expression patterns [15]. Constitutively produced homeostatic chemokines, such as CXCL12, CXCL13, CCL14, CCL19, CCL20, CCL21, CCL25, and CCL27, are responsible for basal leukocyte migration, controlling of leukocyte homing and lymphocyte recirculation in normal conditions [15]. Inflammatory chemokines, such as CXCL8, CXCL10, CCL2, CCL3, CCL4, CCL5, and CCL11, are formed in pathological states after induction by various pro-inflammatory factors, including interleukin 1 (IL-1), tumor necrosis factor alpha (TNF-α), lipopolysaccharides (LPS), and viruses. It results in the accumulation of leukocytes in the site of inflammation and their activation during inflammatory reactions [15]. One of the most important mediators of inflammation is CXCL8, also known as IL-8, which creates together with receptors CXCR1 and CXCR2 a specific signaling pathway, called “CXCL8-CXCR1/2 axis” [15].

Chemokines are the mediators of the immune system, which regulate inflammation, infection, immunological response, and tissue injury reactions, as well as apoptosis and immune cell traffic in host defense [16]. They play also a role in the tissue development and functioning [11,17]. As chemoattractants, they stimulate the migration of mesenchymal, epithelial, and endothelial cells as well. These proteins regulate tissue extravasation and functional modulation of cells [17]. Chemokines regulate hematopoiesis, inducing migration, survival, and other activities of hematopoietic stem and progenitor cells [18]. Chemokines are also essential factors in processes of embryo implantation, placentation, and controlled trophoblast invasion [19].

Among homeostatic chemokines, CXCL12, also known as SDF-1 (stromal derived factor 1) is one of the most primitive and strongly conserved through evolution chemokines, which binds to CXCR4 receptor. They form the most obligatory for life chemokine-receptor axis, CXCL12-CXCR4, an essential player for the development of many organs [11,20]. It was shown that CXCL12 and CXCR4 are crucial for the development of various organs, such as kidney or gastrointestinal tract [21,22]. CXCL12-CXCR4 axis participates also in or lympho- and myelopoiesis [23,24,25]. Interestingly, CXCL12 and CXCR4 are widely expressed in the central nervous system (CNS), mainly on astrocytes and microglia of the normal CNS as well as on neurons of adult brain [26]. They are essential for the proper functions of human neural progenitor cells (hNPCs) [27].

CX_3_CL1, the only known member of the CX_3_C chemokine family, is expressed constitutively by neurons in the brain and spinal cord [28], while in astrocytes it may be upregulated by TNF-α and IL-1β [29]. This chemokine is also expressed on activated endothelial cells, dendritic cells, and intestinal epithelial cells [30,31,32]. CX_3_CR1, the unique receptor for fractalkine, is present on T cells, natural killer (NK) cells, monocytes [33], and microvascular endothelial cells [34].

Therefore, chemokines and their receptors were proposed as the new class of CNS neuromodulators, which can participate in neuronal signaling [35,36,37] and regulate the neuroendocrine system [28]. Moreover, it was demonstrated that chemokines may have important function in cerebellar development [38] and controlling of axon guidance [39], as well as neuroinflammation and synaptic transmission [39].

CXC12-CXCR4 axis is also thought to be the crucial controller of early brain cortex development, proliferation of neurons and progenitor neural cells, and their directed migration [40,41]. The differentiation of neural progenitors is controlled by CXCL12, together with another chemokine CCL2, which are secreted constitutively by blood–brain barrier (BBB) endothelial cells [42]. Furthermore, CXCL12 enhances survival of human progenitor neural cell acting through its receptor CXCR4 [20]. CXCL12 and CXCR4 cooperate during migration of facial motor neurons [43] and regulate interneuron relocation [44]. Apart from their role in the brain development, chemokines and their receptors may also have important functions in processes of learning and memory. mRNA of many receptors for chemokines from CC and CXC groups was shown in neural stem cells in the subventricular zone of the hippocampus, with CXCR4 presenting the highest expression [45]. Interestingly, not all chemokines play a role of chemo-attractants in the development of CNS, some of them may act as inhibitors of neural cells. For example, the CXCL1/CXCR2 axis may arrest the migration and positioning of oligodendrocyte precursors from developing spinal cord ventricular zone to the white matter [46].

Another chemokine function, especially CXCL16, which acts through its unique receptor CXCR6, is mediating in the homeostatic glial-neuronal communication [47]. This chemokine promotes also a neuroprotection from glutamate-induced excitotoxicity [48], mediates endogenous protective mechanisms against ischemic neuronal damage [49], and modulates neurotransmitter release in the hippocampus [50]. Moreover, CXCL16/CXCR6 axis drives microglia/macrophages phenotype in physiological conditions [51].

## 5. Chemokines as Angiogenic Mediators

One of the most important roles of chemokines is their activity as mediators of angiogenesis, a growth and proliferation of new blood vessels from pre-existing vasculature. New blood vessel growth may occur in embryonic development and in response to ischemia under some postnatal physiological conditions, such as menstrual/ovarian cycle, wound repair, or tissue neovascularization. However, this process may also progress uncontrolled, in response to prolonged or excessive inflammation in some pathological processes, e.g., tumor growth, or atherosclerosis. There are a variety of factors which can mediate the angiogenic process, including adhesion molecules, such as vascular cell adhesion molecule-1 (VCAM-1); cytokines from fibroblast growth factor (FGF) family; angiopoietins Ang1 and Ang2; matrix metalloproteinases (MMPs), and certain chemokines [52,53,54].

### 5.1. CXC Chemokines

The crucial role in the regulation of angiogenesis was demonstrated especially for CXC chemokines, which have an ability to act both as pro- or anti-angiogenic factors [55,56,57]. The angiogenic potential and chemotactic properties of these subfamily members depend on the presence of conserved sequence of the three AAs, Glu-Leu-Arg, termed as “ELR motif”, in the N-terminal domain of chemokine molecule [55,58]. Members of CXC ELR (+) chemokine subfamily are chemotactic for neutrophils, although in the absence of preceding inflammation, they can mediate angiogenesis [55]. This group of pro-angiogenic chemokines includes CXCL1 (growth-related oncogene alpha, GROα), CXCL2 (GROβ), CXCL3 (GROγ), CXCL5 (epithelial neutrophil-activating protein-78, ENA-78), CXCL6 (granulocyte chemotactic protein 2, GCP-2), CXCL7 (neutrophil-activating protein-2/NAP-2), and CXCL8 (interleukin 8, IL-8) [59]. Among them, CXCL8 is the prototypic pro-angiogenic chemokine in its promotion of endothelial cell migration, invasion, and proliferation [60].

CXC ELR (+) chemokines may induce angiogenesis through several ways. One of them is the recruitment of pro-angiogenic hematopoietic and immune cells as well as endothelial progenitors to the neo-vascular niche [61]. They may also regulate the functions of endothelial cells by direct activation of cCKRs expressed on these cells, what results in the induction of chemotaxis and tubular morphogenesis. Furthermore, many chemokines and CKRs pathways have cross talk with other pro-angiogenic factors signaling, such as vascular endothelial growth factor (VEGF) and FGF, resulting in differential effects on angiogenesis. Chemokines may also induce angiogenesis via direct interaction between chemokine/chemokine receptor complexes and tyrosine kinase receptors [56].

All human CXC ELR (+) chemokines signal primarily through the receptor CXCR2, although they can also communicate by CXCR1 [62]. CXCR2 was detected in human microvascular endothelial cells and blocking of this receptor by neutralizing antibodies resulted in the lack of angiogenic activity induced by CXC ELR (+) chemokines [63]. These findings indicate that CXCR2 is the receptor responsible for chemotaxis and neovascularization mediated by CXC ELR (+) chemokines.

On the contrary, chemokines ELR (-), which may be induced by interferons (IFN), have angiostatic properties [55,64,65,66]. This group includes CXCL4 (platelet factor-4, PF4), CXCL4L1 (CXCL4 variant), CXCL9 (monokine induced by interferon-γ, MIG), CXCL10 (interferon-γ inducible protein-10, IP-10), CXCL11 (interferon-γ inducible T cell chemoattractant, I-TAC), CXCL12, CXCL13 (B cell attracting chemokine-1, BCA-1), and CXCL14 (breast-and-kidney-expressed chemokine, BRAK) [57,59]. CXC ELR (-) chemokines are the chemoattractants for lymphocytes T and NK cells [59].

Inhibition of angiogenesis may undergo in several ways. It can occur through a positive feedback loop, in which angiostatic chemokines stimulate the recruitment of Th1 lymphocytes and NK cells [67]. Additionally, angiostatic chemokines CXCL9, CXCL10, and CXCL11, activate these cells to secrete interleukin, what leads to increased production of IFN-γ. The regulation of IFN-inducible ELR (-) chemokines expression and induction of Th1-type immune response is a reciprocal process [67]. IFN-γ stimulates the expression of CXCR3 ligands, i.e., CXCL9, CXCL10, and CXCL11, which again increase the recruitment and activation of CXCR3-expressing cells. Moreover, it was demonstrated that the interaction of monocytes with vascular endothelial cells may also induce synergistically the expression of CXCL10 through the activation of specific cell surface molecules and cytokines [68]. In addition, angiostatic chemokines may inhibit angiogenesis through binding to the receptors expressed on endothelial cells, what induces apoptosis or regression of vessels. Anti-angiogenic activity of ELR (-) chemokines may occur also by binding of angiogenic growth factors, what results in the inhibition of their activity.

It was demonstrated that CXCL4, CXCL9, CXCL10, and CXCL11 chemokines may directly stop chemotaxis of endothelial cells and prevent the formation of endothelial tube induced by growth factors [69]. Furthermore, it was shown that CXCR3 receptor, which is expressed on the endothelial cell surface, may bind the anti-angiogenic chemokines CXCL4, CXCL4L1, CXCL9, CXCL10, and CXCL11 [69]. It was demonstrated that CXCR3 may mediate the angiostatic activity of ELR (-) chemokines both in vitro and in vivo in chick chorioallantoic membrane and Matrigel plugs in mice [55,64,70,71,72]. These CXCR3 ligands inhibit microvascular human endothelial cell migration and proliferation in response to various angiogenic factors [57]. Moreover, CXCR3 ligands induced the regression of newly formed cords in vitro and the loss of blood vessels in vivo. These results suggest that the angiostatic function of CXC ELR (-) may result as independent of cell type. The importance of CXC ELR (-) chemokines in maintaining vascular homeostasis may be highlighted by the fact that mice lacking CXCR3 show disproportionate formation of blood vessels during skin wound healing [69,73]. It was also demonstrated that chemokine CXCL10 may induce the dissociation of newly formed vessels and their regression during wound healing, as well as endothelial cell death, whereas antibodies neutralizing both CXCR3 and CXCL10 could inhibit cord dissociation mediated by this chemokine [74].

Interestingly, despite being ELR (-), the aforementioned chemokine CXCL12 may act as pro-angiogenic factor [75]. It was demonstrated that CXCL12 binding to CXCR4 may increase the production of VEGF, which is the major angiogenic stimulator [76]. This receptor has a wide cellular distribution, with expression both on immature and mature hematopoietic cells, including various types of leukocytes (T-lymphocytes, B-lymphocytes, monocytes and macrophages, neutrophils and eosinophils), smooth muscle cell progenitors, endothelial cell precursors, and vascular endothelial cells, as well as neuronal and nerve cells [77,78]. CXCR4^+^ cells are able to participate in angiogenesis in a direct way, by binding CXCL12 to its receptor on endothelial cells and by recruiting endothelial progenitor cells [79]. Additionally, CXCL12 may indirectly stimulate angiogenesis, by inducing secretion of pro-angiogenic VEGF, CXCL8, and CXCL1 by endothelial cells and leukocytes expressing CXCR4 [80].

### 5.2. CX_3_C Chemokines

Apart from proangiogenic CXC ELR (+) and angiostatic CXC ELR (-), there are certain chemokines from other groups, namely CC and CX_3_C, that may influence processes of growth and proliferation of new blood vessels. It was shown that fractalkine, the only known member of CX_3_C chemokine family and a chemoattractant for monocytes, NK cells, and lymphocytes, might also act as promoter of angiogenesis. Its angiogenic activity is mediated through several different mechanisms. Both CX_3_CL1 and its receptor, CX_3_CR1 may be expressed on endothelial cells [30,81]. Moreover, CX_3_CL1 may stimulate BM-derived monocytes expressing fractalkine receptor CX_3_CR1 to differentiate into the smooth muscle-like cells during healing of blood vessel walls after their injury [80]. The efficient interactions between CX_3_CL1 and its receptor CX_3_CR1 are required for microvessel budding, their maturation, and vascular structural integrity [82]. Moreover, it was demonstrated that disturbed interaction of CX_3_CL1 and CX_3_CR1 resulted in the growth of smaller, poorly developed, leaky, and hemorrhagic microvessels in experimental models of neovascularization [83]. It was revealed that recombinant CX_3_CL1 may induce the proliferation, migration, and formation of endothelial tube in vitro as well as stimulate angiogenesis in vivo [81]. In addition, fractalkine-induced angiogenesis is associated with phosphorylation of some enzymes, such as extracellular-signal-regulated kinase (ERK), Akt, and endothelial nitric oxide (NO) synthase (eNOS). In ischemic conditions, eNOS stimulates the synthesis of NO, which directly promotes angiogenic processes. NO controls the expression of proangiogenic factors such as VEGF, FGFs angiopoietins, and genes involved in extracellular matrix metabolism, including MMPs [84]. It was demonstrated that the activation of both signal pathways via the receptor CX_3_CR1 is required for full CX_3_CL1 angiogenic activity [81].

### 5.3. CC Chemokines

Furthermore, it was shown that some members of the largest chemokine subfamily, CC chemokines, may also influence angiogenesis. This group includes at least 27 members which are chemotactic for multiple types of cells, including monocytes, dendritic cells (DCs), eosinophils, basophils, lymphocytes, and NK cells [15]. They are involved in the immunological surveillance, lymphocyte homing, and inflammatory response during infection and tissue injury. Some of these chemokines are expressed constitutively under homeostatic conditions, while other are induced by various inflammatory stimuli [85]. Examples of CC chemokines implicated in angiogenesis are CCL2 (monocyte chemoattractant protein-1, MCP-1) [57], CCL11 (eotaxin) [86], CCL16 (liver-expressed chemokine, LEC or monotactin-1, MTN-1) [87], and CCL21 (secondary lymphoid-tissue chemokine, SLC) [88].

Although primarily recognized as monocyte chemoattractant, CCL2 may act directly on endothelial cells as a stimulator of angiogenesis [89]. It was demonstrated that endothelial cells express CCR2, the receptor for CCL2, and in response to CCL2 they exhibit chemotaxis and endothelial tube formation in vitro [90]. CCL2 may also have an indirect influence on tumoral angiogenesis by attracting tumor-associated macrophages (TAM), which further secrete other pro-angiogenic factors such as VEGF, platelet-derived growth factor (PDGF), transforming growth factor (TGF)-β, and chemokines (CXCL8), as well as proteolytic enzymes MMP-2 and MMP-9 [91]. Moreover, CCL2-induced tube formation was highly dependent on membrane type 1-matrix metalloproteinase (MT-1 MMP) activity. It was revealed that CCL2 may regulate the expression of MT1-MMP in endothelial cells during angiogenesis, increasing significantly the surface expression of this metalloproteinase, its clustering, activity, and function in human endothelial cells [92].

CCL11, a chemoattractant for eosinophils, has pro-angiogenic properties as well. CCR3, the receptor for CCL11, is expressed mainly on the surface of immune cells. This chemokine may directly induce chemotaxis of endothelial cells in vitro, through binding to CCR3 [86]. In addition, CCL11 may induce the proliferation of endothelial cells. It was demonstrated in a rodent model of angiogenesis that CCL11 directly caused microvessel sprouting, what occurred earlier and with greater effects than after stimulation with VEGF [93]. Additionally, an indirect pro-angiogenic activity of CCL11 and formation of blood vessels was also observed in vivo. Stimulation by CCL11 in mice resulted in the infiltration of eosinophils, which release other pro-angiogenic factors [86].

CCL16 is a chemokine predominantly expressed in the liver [94], but also found in the blood and brain [95] and is known to play important roles in the immune response and angiogenesis. By binding to CCR1, CCL16 activates an angiogenic program in vascular endothelial cells and induces migration of endothelial cells in a dose-dependent manner [87]. Moreover, CCL16 promotes endothelial differentiation into capillary-like structures and formation of endothelial tube in vitro. Interestingly, CCL16 signaling has cross talk with other pro-angiogenic pathways through increased basal production of CXCL8 and CCL2, and primes endothelial cells to mitogen signals by VEGF [87].

Another CC chemokine related to angiogenesis is CCL21, which is also known as 6C-kine because of six conserved cysteine residues instead of the usual four conserved ones typical to this subfamily of chemokines [96]. CCL21 belongs to chemokines expressed constitutively under physiological homeostatic conditions in cells. Its main role is to assist with basal leukocyte migration [97,98]. Moreover, CCL21 is expressed by endothelial cells within lymph nodes and in lymphoid neogenesis may act as a growth factor for these cells [88,99]. Human CCL21 binds to the chemokine receptor CCR7, although it was suggested that this chemokine might also bind to endothelial cells via extracellular matrix components such as GAGs [88]. Interestingly, it was demonstrated that murine 6C-kine, but not human CCL21, may also bind to CXCR3, which is a CXC chemokine receptor, playing the role of an anti-angiogenic factor. After the treatment with murine 6C-kine the vessel density was significantly reduced in the mice model when compared with control animals [100].

## 6. Role of Chemokines in Carcinogenesis

### 6.1. Cancer-Related Inflammation

Chemokines are also significant players in the pathogenesis of cancer and cancer-related inflammation [101], which determine the composition of tumor stroma. What is important, chronic inflammation is an essential component of the tumor microenvironment and one of the hallmarks of cancer [102,103]. Being important factors both in inflammation and in cancer, chemokines are the link between these pathways [104]. Chemokines also stimulate directly cancer cells in numerous types of malignancies and in all the stages of malignant disease [105,106], participating in tumor growth, proliferation, invasion, and migration to metastatic sites. Within tumor tissue, chemokines and their receptors are expressed also on stromal cells, what leads to leukocyte recruitment and activation, angiogenesis, and cancer cell migration. In the pathogenesis of cancer, the expression of chemokines and their receptors is often controlled by transcription factors and oncogenic pathways [104].

Leukocyte recruitment to the tumor site is mediated by inflammatory chemokines from the CXC subfamily (CXCL1, -2, -5, -6, and -8) as well as from the CC group (CCL2, -3, and -5), which attract leukocytes: neutrophils bearing receptors CXCR2 and CCR2-positive monocytes. These inflammatory cells differentiate into tumor associated neutrophils (TANs) and TAMs, respectively [107,108,109,110]. Interestingly, recruitment of these inflammatory cells within tumor may exert pro- or anti-tumoral effects. The interaction between CXCL16 and its receptor CXCR6 also induce the polarization of macrophages in solid tumors to a pro-tumoral phenotype [51,111].

### 6.2. Tumoral Neo-Angiogenesis

Chemokines from CXC and CC subfamilies may influence not only normal angiogenesis triggered by ischemic stimuli, but also inflammatory-mediated neoangiogenesis within tumor tissues, what is a necessary process for further tumor development, progression, and metastatic spreading [85,112]. Endothelial cell survival and tumor angiogenesis are promoted by CC chemokines: CCL2, -11, -16, and -18, as well as by proangiogenic CXC-ELR (+) chemokine CXCL8 [113,114]. CXCL16-CXCR6 axis also induces angiogenesis in autocrine signaling pathway involving hypoxia-inducible factor 1-alpha (HIF-1α) [115]. Another example of proangiogenic chemokine activity are CCL2 and CXCL12, which stimulate angiogenesis and inhibit apoptosis. These processes occur by direct binding of given chemokine to a respective chemokine receptor, CCR2 and CXCR4, which are expressed by tumor vessels, or by promoting the recruitment of leukocytes [116,117].

### 6.3. Tumor Microenvironment

Cancer cells may gain the ability to synthesize and express cCKRs and respond to chemokines with pro-tumoral activity [106]. These chemokines may be produced both by tumor cells and infiltrating leukocytes (TANs and TAMs) or cancer-associated fibroblasts (CAF) [118,119]. Chemokines mediate the host-response to cancer by directing leukocytes towards the tumor microenvironment (TME). Tumor cells are able to use various chemokine/receptor axes to take control over the host cells’ signaling and regulatory mechanisms responsible for the synthesis of various growth factors [120,121]. These mechanisms may be used by tumor cells to maintain their own growth and proliferation. CXCL8 also attracts chemotactic neutrophils to tumor sites and stimulates them to secrete various growth factors [101,122]. Moreover, tumor cell may also acquire the capacity of expressing chemokine receptors related to tumor growth [123]. Overexpression of these receptors leads to the establishment of a feedback loop, which allows cancer cells to divide and proliferate under the influence of chemokines available in the TME and the promotion of tumor growth [124]. Furthermore, through binding of CKRs expressed by tumor cells, chemokines activate various signaling pathways, such as PI3K/AKT/NF-κB and MAPK/ERK, what directly promotes proliferation of cancer cell proliferation [78,125,126]. Chemokines may also exert their pro-tumoral activity by influence on the equilibrium between anti- and pro-apoptotic factors as well as by blocking cancer cell apoptosis [120,127].

An important role in the maintenance of tumor microenvironment and the progression of tumors is attributed to TAMs [128], the most frequent component of the leukocyte infiltrate within tumors [129]. Chemokines are responsible for TAM polarization into different phenotypes: M1 or M2 [130] depending on the cytokines expression in TME and their stimulatory properties. Pro-inflammatory type M1 macrophages are classically activated by IFN-γ and TNF-α and play important roles in the direct host defense mechanisms, such as phagocytosis and secretion of pro-inflammatory cytokines. Moreover, M1 cells may exert anti-tumoral activity. They synthesize CXCL9 and CXCL10, the ligands of chemokine receptor CXCR3, whose presence is required for M1 macrophage generation [131]. On the contrary, alternatively activated anti-inflammatory M2 macrophages are induced by IL-4, IL-13, and TGF-β, and may support tumor growth [132]. M2 polarization of macrophages may be also promoted by chemokines CCL2, CCL17 (thymus and activation-regulated chemokine, TARC), and CCL22 (macrophage-derived chemokine, MDC) [91]. M2 cells are involved in immunosuppression, tissue repair, angiogenesis, and tumor promotion [91,133].

### 6.4. Tumor Growth

Chemokines can promote local tumor invasion, and metastatic spreading of cancer cells to regional lymph nodes and distant tissues [17,134]. Since cancer metastasis and leukocyte trafficking are the examples of chemokine-directed cell migration through blood vessels to distant organs, it indicates the significant role of chemokines in malignant tumor progression [135]. CXCL12-CXCR4 axis is one of the most important signaling pathways, which provides surviving signals to various cancer cells, including hepatocellular carcinoma, ovarian cancer, or chronic leukemia [78,136,137]. Moreover, inhibition of CXCL12-CXCR4 axis exerts antitumoral effects, such as induction of malignant cells apoptosis [138].

CXCL8-CXCR1/2 axis is also an important chemokine/receptor system, that regulates proliferation of cancer cells, mediates initiation and progression of various cancers, and is associated with early relapse and poor prognosis [139]. CXCL8 may be secreted both by tumor cells and TAMs, what facilitates cancer cells’ survival and induces resistance to the chemotherapy [140]. CXCL8 also activates MMPs, which degrade the basement membrane and extracellular matrix, promoting tumor cell migration, infiltration, and distant metastasis as well as angiogenesis within the tumor [141].

### 6.5. Metastasis

Metastatic potential of cancer cells and their migration to distant organs may be promoted by the presence of chemokine receptors [17]. In cancer spreading, CXCR4 is the main cCKR, whose expression enables cancer cells to migrate and metastasize into the organs that secrete high levels of its ligand, CXCL12 [106,142]. Another chemokine receptor, involved in metastasis of non-small cell lung cancer, is CCR7 mediating the migration of tumor cells to lymphatic nodes, where its ligands are produced, CCL19 and CCL21 [17,143]. CCR10/CCL27 axis also facilitates the adhesion and survival of malignant cells during metastatic dissemination of melanoma [144]. Furthermore, bone metastases in prostate cancer are supported by chemokine CXCL13 and its receptor CXCR5 [145]. In addition, CCL28 promotes breast cancer growth and metastasis spreading through MAPK/ERK pathway, MAPK-mediated cellular anti-apoptosis and pro-metastasis [146].

## 7. Gliomas: Malignant Tumors of Central Nervous System

The primary tumors of the CNS represent 2% of all cancer cases worldwide and include malignant neoplasms of brain, spinal cord, meninges, cranial nerves, and other parts of CNS, according to the 10th Revision of International Classification of Diseases [147]. Primary CNS tumors histologically constitute also a highly heterogeneous group of neoplasms, with different courses of the disease and the patients’ prognoses. Moreover, the frequency of specific histological types of CNS tumors varies between different age groups. Generally, intracranial tumors have a predilection for adults and global age-standardized incidence rate per 100,000 population of CNS in population under 20 years is approximately three, while in age group over 65 years it is almost five times higher [147]. Brain tumors are one of the leading causes of cancer-related death worldwide, with similar mortality and incidence rates [148].

CNS tumors represent also a range of low- and high-grade states, which may be classified as grade I, II, III, or IV, according to the 2016 World Health Organization (WHO) Classification of Tumors of the Central Nervous System [149]. Whereas meningiomas, brain tumors derived from meningothelial cells, account for approximately 30% of all malignant tumors of CNS, the most frequent malignancies are gliomas. They constitute over 40% of all primary malignant CNS tumors and are of neuroepithelial origin, which derive from the cellular elements of the glia, i.e., astrocytes and oligodendrocytes.

Low-grade infiltrating gliomas (LGG) of the cerebral hemispheres are currently assigned a WHO grade of II. LGGs are a diverse group of primary brain tumors and include astrocytomas, oligodendrogliomas, and oligoastrocytomas. They often develop in young, otherwise healthy patients and generally have a less invasive course with longer-term survival in comparison with high-grade gliomas (HGG). Unfortunately, grade II gliomas have the potential to progress to more aggressive grade III and IV malignant tumors. Grade III glial tumors comprise anaplastic astrocytoma, mixed anaplastic oligoastrocytoma, and anaplastic oligodendroglioma, while glioblastoma (GBM) represents WHO grade IV. In adults, high-grade anaplastic astrocytomas and GBM constitute about 55% of all neuroepithelial tumors [150]. Moreover, IV grade GBM is characterized with the worst prognosis among all CNS tumors.

## 8. Angiogenic Factors in Gliomas: Transition to High-grade Tumors

Primary GBM (PrGBM) are the tumors developing de novo, mainly in older patients (typically over 60 years of age), without a pre-existing lower-grade glioma, and they account for approximately 90% of all GBM tumors. Secondary GBM tumors (ScGBM) (about 10% of cases) typically occur in younger patients (before 45 years of age) and arise from already existing tumors, especially from grade II or III astrocytomas or mixed gliomas bearing certain genetic mutations [149]. It was shown that the progress of LGG into their malignant high-grade counterparts may occur within 5 years after its gross-total resection (<1 cm residual tumor) even in about 50% of patients [151,152].

Interestingly, despite histological similarity between PrGBM and ScGBM, these tumors exhibit distinct genetic alterations. Whereas isocitrate dehydrogenase 1/2 (IDH)1/2 mutations are frequent (>80%) in ScGBMs that have progressed from low-grade or anaplastic astrocytomas, PrGBM bearing this mutations account for less than <5%. Other alterations significantly more frequent in ScGBM include *TP53* mutations, loss of heterozygosity (LOH) 19q, and LOH 22q [153,154]. On the contrary, genetic alterations more characteristic for PrGBM than ScGBM are LOH 10p, amplification of epidermal growth factor (EGFR), and *PTEN* mutations [153,155]. Therefore, PrGBM and ScGBM can be considered as two different diseases [156], although they have similarly unfavorable outcome, with a median survival shorter than 1 year after diagnosis.

It was primarily supposed that gliomas do not metastasize to distant organs and their growth is restricted to CNS. In fact, malignant gliomas are usually locally invasive tumors, although in some cases such extra-neural metastases (ENM) may occur later in the course of the disease (median of 2 years) [157]. Therefore, ENMs of glioblastoma multiforme are rare due to the short survival of the patients. Although ENMs generally appear after craniotomy, what may suggest rather their iatrogenic origin, some spontaneous metastases have also been reported. The incidence of these metastases from primary intra-cranial malignant gliomas is estimated at less than 2% of all cases. Most frequent ENMs site include bones, lung, lymph nodes, liver, and neck [158].

Although several genetic alterations, including mutations, activation of oncogenes, loss of telomerase and induction of aneuploidy, as well as some molecular changes and epigenetic alterations have been reported as the factors affecting the development of CNS tumors, together with gliomas [149,159], the precise etiology of these malignancies is still unclear. It seems that some cytokines, chemokines, and their receptors may play a regulatory role over certain steps of gliomagenesis, such as tumor proliferation, evasion of cell apoptosis, migration, and angiogenesis.

Tumoral angiogenesis differs significantly from described above physiological angiogenesis and is characterized by aberrant blood flow, abnormal vascular structure, higher vessel permeability, and altered interactions between endothelial cells and pericytes. These atypical features of the tumor vasculature may result from the altered balance between the expression of various pro- and antiangiogenic factors, such as cytokines and chemokines, as well as the status of tumor suppressor protein p53, which can regulate key angiogenic cytokines and inhibitors. Moreover, expression of these factors may vary between different tumor types [160].

As it was mentioned earlier, some LGGs, such as diffuse astrocytomas and oligodendrogliomas, may have the potential to transform into infiltrative malignant HGGs. Generally, LGGs are characterized by next to no neovascularization, a normal process that occurs by way of ischemic stimuli, resulting in linear growth of tumors [161]. On the contrary, HGGs, including high-grade astrocytoma, oligodendrogliomas, and ependymomas, are characterized by abundant hypervascularization which provides the glioma with blood supply sufficient for exponential growth [162]. Maximal vessel density is observed in GBM, which belongs to the most vascularized tumors [163]. The explanation for this phenomenon might be that low-grade CNS tumors come across so-called “angiogenic switch”, which is an induction of a tumor vasculature, resulting in a rapid formation of new blood vessels, thus promoting tumor growth and invasiveness [164]. This switch may allow quick progression and malignant transformation toward high-grade and may occur at any tumor stage, depending on type of the tumor and its microenvironment [165,166].

Stimulation of the angiogenic switch is associated with increased expression of pro-angiogenic genes, induced by physiological stimuli, such as hypoxia within tumor tissue resulting from increased tissue mass [167]. In this machinery, low oxygen states induce glioma stem cells to release signaling mediators, such as VEGF, PDGF, and HIF-1α [167]. It was demonstrated that HGGs express some transcriptional alterations in angiogenesis-associated factors, such as VEGF, fibroblast growth factor (FGF), and epidermal growth factor (EGF), that correlate with neovascularization in human GBM samples [168], and the upregulation of these genes may also play a role in activating the angiogenic switch.

## 9. Deregulated Chemokine Network Characteristic for Gliomas

As it was described above, for their formation and progression, malignant tumors may employ various chemokines and their receptors. CNS tumors, including gliomas, are no exception in this regard. Various pathophysiological mechanisms in gliomas may be regulated by cytokines, including chemokines [169]. It seems that chemokines and their receptors form a kind of communication network that directly participates in critical processes of glioma growth, thus accounting for the increased malignancy of glioma.

### 9.1. CXCL8-CXCR1/2 Axis

The CXCL8-CXCR1/2 axis belongs to the most important and the best recognized regulatory factors in the development of CNS tumors. CXCL8 is one of the inflammatory ELR (+) chemokines, which is not only a critical regulator of normal CNS function and development, but plays an important role in many CNS disorders, including malignant tumors. Similarly to other malignancies, an upregulation of CXCL8 in tumor cells was observed also in gliomas [139]. Activity of this chemokine is related with various steps of gliomagenesis: development, progression, and recurrence (Figure 2).

In human gliomas, CXCL8 is expressed and secreted at high levels by tumor cell both in vitro and in vivo [170]. Expression of CXCL8 mRNA and protein were detected in grades II, III, and IV astrocytomas and anaplastic oligodendroglioma [170]. CXCL8 protein expression and its promoter activity in glioblastoma cells may be induced by other pro-tumoral cytokines, such as IL-6 [171] or by anti-apoptotic proteins from Bcl-2 family, such as Bcl-xL [172]. An increased ratio of anti-apoptotic Bcl-2 family proteins, particularly Bcl-2 and Bcl-xL, to pro-apoptotic proteins Bax and Bcl-xs results in the increased tumor cell survival mediated by CXCL8 [173]. These actions of CXCL8 are additionally increased by the expression of proteolytic enzymes, especially MMP-2 and -9, which also contribute to the enhanced glioma cell invasion [173]. Moreover, it was revealed that this upregulation of CXCL8 in glioblastoma cells induced by Bcl-xL, is mediated through a nuclear factor-kappa B (NF-kB)-dependent mechanism [172].

Importance of upregulated CXCL8 level in glioma was considered by many researchers. It was demonstrated that CXCL8 expression levels in tumor samples from GBM patients were significantly higher than in normal brain tissue and correlated with tumor progression [174]. GBM patients with high levels of CXCL8 displayed shorter disease-free and overall survival, showing that this chemokine may be a prognostic factor of survival in glioblastoma patients. It was also revealed that glioma grade strongly correlates with the levels of CXCL8 [170,171]. Moreover, CXCL8 was detected in the cyst fluid of primary astrocytic tumors [175]. Serum concentrations of this chemokine and PDGF were significantly higher in patients with GBM, compared to the healthy subjects [176]. Furthermore, cerebrospinal fluid (CSF) levels of CXCL8 were significantly elevated in patients with CNS tumors as compared to non-tumoral control group [177].

Like other malignant solid tumors, GBMs are characterized by biological complexity. Apart from neoplastic glioma cells, these tumors are also composed of many non-neoplastic cells: neurons, glia, macrophages, endothelial cells, lymphocytes, and neutrophils, which create tumor stroma. Each of these cell types may secrete CXCL8, thus contributing to gliomagenesis. Specifically, high amounts of this chemokine may be produced by macrophages [178]. Within normal brain tissue, there is a resident population of macrophages, that are resting in the perivascular space. In the course of various CNS diseases, including HGG tumors, these macrophages become activated and may migrate to the region involved in the pathological process, where they participate in the inflammatory response. While in WHO grade II astrocytomas macrophages are not especially frequent, the number of these cells grows with increasing histologic grade of the tumor, being the highest in GBM [179]. Moreover, it was demonstrated that there is a strong positive correlation between level of CXCL8 and the number of macrophages in astrocytic neoplasms. In addition, it was estimated that the density of macrophage infiltrates correlated with the degree of microvascular proliferation [179].

Another process important for GBM proliferation, cell migration, and invasion, in which CXCL8 participates, is vascular mimicry (VM). VM is a novel, alternative neovascularization mechanism, defined as the generation de novo of microvascular channels by aggressive, metastatic, and genetically deregulated tumor cells, used not only by gliomas, but also by other malignant tumors [180]. Formation of microvascular channels is a manner in which malignant tumors acquire a blood supply for their growth and hematogenous dissemination, but without participation by endothelial cells. VM is an ability of tumor cells to obtain endothelial-like properties and to create vascular structures embedded in ECM, which contain plasma and blood cells. These structures form a kind of vascular-patterned networks, which can only imitate normal host’s endothelial blood vessels. The result of this process is that GBM vessels are disorganized and destabilized structures with high permeability and tortuous shape, which present also abnormal endothelial and pericyte coverage [181,182]. As VM is independent of angiogenesis, malignant tumors, such as GBM, which use this mechanism of vascularization, may display therapeutic resistance to antiangiogenic therapies (AAT), especially CXCR2-expressing tumor cells [183]. What is especially important, some AAT-treated patients may even show higher rates of relapse and potent refractoriness of tumor due to AAT-mediated enhancement in VM [184]. Interestingly, anti-angiogenic treatment resulted in the increased number of CXCR2-positive GBM stem cells with endothelial-like phenotypes, what may confirm that GBM employ this alternative neovascularization mechanism. On the other hand, a reduction of CXCR2 expression in tumor cells led to the inhibition of tumor growth and development of incomplete VM structures in the animal GBM models [184].

Furthermore, it was shown that CXCL8 may promote the epithelial–mesenchymal transition (EMT) [174]. EMT is an essential process in the malignant cell proliferation, migration, and invasion, including gliomas, where epithelial tumor cells lose their basal-apical polarity and become motile mesenchymal cells [185]. It was demonstrated that CXCL8 participates in this cellular process by activating the JAK/STAT1/HIF-1α/Snail signaling pathway in glioblastoma cells, thus promoting glioma progression [174].

Although this chemokine was shown to induce four main processes of tumorigenesis, including proliferation, chemotaxis, survival, and protease activation, the key contribution of CXCL8 and its receptors in gliomagenesis is related to the promotion of angiogenesis. This chemokine exerts its pleiotropic effects through G-protein coupled, conventional chemokine receptors CXCR1 and CXCR2 [170]. The proangiogenic activity of CXCL8 in gliomas is mainly related to CXCR2, which binds to all of the CXC ELR (+) chemokines. However, CXCR1 may also contribute to pro-angiogenic functions of this chemokine, through independent small GTPase activity. CXCL8 high affinity interaction with endothelial CXCR2 triggers cells proliferation [170], and consecutive migration of endothelial cells, which form a basic vessel structure with the central lumen [186]. The influence of CXCL8 on tumor angiogenesis is regulated by Bcl-xl protein. Overexpression of Bcl-xl in human glioblastoma cells induced an increased expression of CXCL8, both at the protein and mRNA levels, and an enhanced CXCL8 promoter activity [187]. In addition, human glioblastoma cell lines induced by Bcl-xl protein showed increased in vitro endothelial cell functions, such as morphogenesis and proliferation, as well as enhanced in vivo formation of vessels in angiogenesis models [187].

Signaling of CXCR2 receptor is strongly activated in the microenvironment of GBM and responsible for tumor neovascularization and metastasis [139]. Similarly to CXCL8, CXCR2 expression was significantly higher in HGG tumors when compared with normal tissues [188]. Moreover, upregulation of this receptor correlated with poor prognosis and recurrence in human glioma [188]. The expression of CXCR2 was also significantly related to high grades of glioma and the recurrence of tumors. In addition, blocking of this signaling by anti-CXCR2 antibody or by CXCR2 inhibitor resulted in suppressed glioma growth and cell migration [188,189]. Highly upregulated CXCL8-CXCR2 axis was identified in HGG tumors resistant to anti-angiogenic therapy [184].

It was recently revealed that enhanced CXCL8 expression in the tumor microenvironment might also contribute to the local and systemic immunosuppression induced by glioblastoma [190], which allows GBMs to evade host immunosurveillance and may create a significant barrier to glioma immunotherapy. GBM-related systemic immunosuppression is associated with the accumulation of immunosuppressive leukocytes such as regulatory T-cells (Tregs) and myeloid-derived suppressor cells (MDSCs) [191,192]. MDSCs, a subpopulation of leukocytes derived from monocytes, are a critical component of this immunosuppression, which may inhibit the proliferation and activation of T lymphocytes. In non-cancer patients, MDSCs are present at low baseline levels, preventing autoimmune reactions and moderating inflammation [193,194]. It was demonstrated that CXCL8 expression by GBM may influence the trafficking of MDSCs into the tumor environment by acting on the CXCR2 receptor [195]. By co-culturing with human GBM cell lines in human glioma-conditioned media, normal monocytes could transform to MDSC [190]. It was also demonstrated that an array of cytokines and chemokines, including CXCL8, CCL2, interleukin-6, and macrophage migration inhibitory factor (MIF), may contribute to MDSC generation [190].

Interestingly, it was revealed that CXCL8 secretion by GBM cells is induced by the necrotic cells existing within tumor tissue [196]. The presence of necrosis is presumed as one of the remarkable histopathological hallmarks of GBM. Moreover, these necrotic tissues may also influence tumor growth and metastasis through inducing chemokines with pro-tumoral activity. Incubation of GBM cell line with necrotic cells resulted in the significant enhancement of the migration and invasion of tumor cells in a dose-dependent manner. Necrotic cells induced the activation of transcription factors, such as NF-κB and activator protein 1 (AP-1), as well as their binding to the CXCL8 promoter. Additionally, immunohistochemical and immunofluorescence examination revealed that CXCL8 positive cells were distributed mainly in the perinecrotic region of human GBM tissues. This CXCL8 upregulation via NF-κB/AP-1 mediated GBM proliferation, what suggests that there is a kind of signaling network between necrotic tissues and tumor [196].

### 9.2. CXCL12-CXCR4 Axis

Another chemokine that plays a crucial role in the growth and proliferation of gliomas is CXCL12, whose activity is mediated through conventional chemokine receptor CXCR4 [197]. Since both CXCL12 and CXCR4 are highly expressed in normal brain, where CXCL12-CXCR4 axis plays an important role in the CNS development, the involvement of this chemokine-receptor axis in glioblastoma may be considered as an example of tumor cells “hijacking” of physiological processes in CNS [198]. CXCR4 and CXCL12 are the potent regulators of glioma stem cell proliferation.

Upregulated expression of CXCL12 and its receptor CXCR4 was observed in various types of malignant CNS tumors, including low-grade oligodendrogliomas, oligoastrocytomas [199], and astrogliomas [200] as well as high-grade glioblastomas [201]. Within human glioblastoma tissues, an increased expression of CXCL12 and CXCR4 was localized not only in the regions of tumor necrosis and apoptosis, but also in the areas characterized with high microvessel density, what indicates the importance of this chemokine in the development of these tumors [201,202]. Moreover, it was shown that CXCL12-CXCR4 axis is involved in the brain infiltration by glioma cells. The enhanced expression of both receptor and ligand was observed especially at the invasive tumor’s edge in glioblastoma [203].

It was demonstrated that CXCR4 is a major chemokine receptor on glioma cells [204]. Among various chemokine receptors and their ligands tested, mRNA expression of both CXCL12 and CXCR4 were identified in the highest percentage of human astrocytic tumor samples [202,204]. Marked co-localization of CXCR4 and CXCL12 in tumor cells was also shown in immunohistochemical examination of tumor samples, they were manifested predominantly in pseudopalisading in the areas of perinecrosis and microcystic degeneration [204]. This formation of pseudopalisading necrosis area, often observed in glioblastoma, is thought to be a result of the migration of tumor cells out of the hypoxia zone [205]. It was also observed that CXCR4 is highly expressed in glioma progenitor cells, whereas its ligand CXCL12 promotes a specific proliferative response in these cells [206].

CXCL12-CXCR4 signaling pathway mediates cellular invasion in glioma. Interestingly, it was revealed that CXCL12 may induce a significant increase of DNA synthesis in primary human glioblastoma cell cultures and chemotaxis in a glioblastoma cell line [204], what proves that this chemokine contributes not only to chemotactic spread of tumor cells, but also induces proliferation of malignant cells. Moreover, the protein expression of CXCR4 increased with the tumor grade and was significantly higher in gliomas of grade III and IV than in gliomas of grade II [201,207]. Such increased CXCL12 expression was an unfavorable predictor of tumor progression in LGGs [208]. In addition, endothelial cells surrounding the tumor tissue in LGGs also express CXCL12, what suggests that this chemokine may contribute to the dissemination of tumor cells [208].

It was shown that pattern of CXCL12 and CXCR4 expression may be also a predictive factor of glioma recurrence after its total resection. Using immunohistochemical techniques in glioma tissues obtained in total tumor resection, it was demonstrated that CXCL12 was expressed mainly in vascular endothelial cells, while CXCR4 immunostaining was observed mainly in tumor cells [209].

Although extensive angiogenesis is one of the key hallmarks of GBM, it is a complex feature of tumor tissues that changes over time. CXCL12 and receptor CXCR4 participate also in this tumoral angiogenesis. As it was detailed above, despite being ELR (-) chemokine, CXCL12 acts as pro-angiogenic factor and stimulates VEGF expression [210], which could be a potential explanation for enhanced invasiveness of brain tumor. Moreover, CXCL12 and its receptor CXCR4 have been shown to promote VEGF production mediated by glioma stem cells and tumor angiogenesis via PI3K/AKT signaling [211].

It was shown that CXCL12-CXCR4 axis generates an amplification loop which is influenced by hypoxia: CXCL12 upregulates VEGF-A production, which, in turn, further enhances expression of CXCR4 [212]. Furthermore, in ischemic endothelial cells, CXCL12 gene expression in tumor cells is controlled by HIF-1α in direct proportion to reduced oxygen level [210]. Hypoxic conditions, often present in brain tumors, lead to the enhanced expression of HIF-1α, what further stimulates CXCL12 expression in tumor cells, affects the spreading of tumor cells by CXCR4 receptor binding in endothelial cells [13], and upregulates the expression of VEGF, increasing angiogenesis. Apart from angiogenesis, CXCL12 seems to induce vasculogenesis in GBM as well [13]. These actions are also induced by the enhanced production of other cytokines like CXCL8 or CCL2, which are regulated by the MAPK pathway [200].

Pro-angiogenic properties of CXCL12 may be also associated with above described “angiogenic switch” of CNS tumors, which allows for recurrence of glioma, its quick progression and malignant transformation toward higher grades. It was proposed that some changes in the expression of angiogenesis-related genes or angiogenic factors must occur between initial diagnosis of GBM and its recurrence after the treatment. These initial-recurrence expression alterations include VEGF-A, its receptors VEGFR2 and VEGFR1, HIF1α, and urokinase plasminogen activator (uPA) as well as CXCL12 and CXCR4, the changes were largely consistent between RNA and protein expression. It was assessed that at GBM recurrence, the expressions of CXCR4 and CXCL12 were increased, while expressions of HIF1α and VEGFR2 were decreased. Recurrence of GBM after chemo-radiation is associated with a switch of angiogenic pattern from VEGFR2-HIF1α to CXCL12-CXCR4 pathway [213]. These results confirm that the recurrence of glioblastoma after combined radio-chemotherapy is associated with an angiogenic switch in the pattern of chemokine expression to proangiogenic and protumoral CXCL12-CXCR4 pathway (Figure 3).

### 9.3. XCL16-CXCR6 Axis

Within normal brain, chemokine CXCL16 is expressed constitutively and plays a neuroprotective role against ischemia [51]. CXCL16 expression in normal brain is low and generally restricted to brain vascular endothelial cells [214]. In GBM, CXCL16 may be an important factor in the modulation of microglia cell activity and their phenotype, as well as in the progression of the tumor. CXCL16 is highly expressed in human gliomas, where both mRNA and protein are upregulated by TNFα and IFNγ [214]. It was reported that CXCL16 released by glioma cells may drive the polarization of microglia macrophages, called glioma associated microglia (GAMs) toward the anti-inflammatory/pro-tumoral phenotype [51].

CXCR6, the only known receptor for CXCL16, is widely expressed in the brain, including microglia, acting as an endogenous protective factor against excitotoxic neuronal damage. The presence of CXCR6 in GBM is associated with glioma-stem cells [215,216]. CXCL16 induces migration and invasion of glial precursor cells via CXCR6 [217]. Overexpression of CXCL16-CXCR6 axis was found in glial tumors [215]. CXCL16/CXCR6 signaling acts directly on human primary GBM cells and mouse glioma cells, promoting tumor cell growth, invasion, and migration [51] (Figure 4).

### 9.4. CX_3_CL1-CX_3_CR1 Axis

CX_3_CL1 and CX_3_CR1 constitute the next chemokine ligand/receptor axis that plays an important role in glioma development [218]. Both these molecules are constitutively expressed by various CNS cell types, including monocytes, NK cells, T cells, dendritic cells, as well as neurons and astrocytes, where CX_3_CL1 controls neuronal survival and neurotransmission [219]. CX_3_CL1 is a transmembrane protein, which may function as an adhesion molecule as well as a chemokine when cleaved by specific proteases [220].

Membrane expression of the chemokine and its receptor was shown on human glioma cells in vitro [221]. Accumulation of soluble CX_3_CL1 was observed in the supernatants from glioma cell lines, indicating that the chemokine is constitutively released also by these tumors. Inhibition of endogenous CX_3_CL1 using a neutralizing monoclonal antibody resulted in marked latency of tumor cell aggregation and increased glioma invasiveness [221]. Moreover, the expression of CX_3_CL1 in glioma cell lines, both on mRNA and protein level, was decreased by the treatment with TGF-β, a key regulator of glioma cell invasiveness.

Furthermore, analysis of CX_3_CL1 and CX_3_CR1 expression in human glioma surgical samples with different histological degrees demonstrated that both ligand and receptor were present in most specimens, at the mRNA and protein levels [218]. Interestingly, the highest CX_3_CL1 expression levels were found in grades III–IV tumors: oligodendrogliomas, anaplastic astrocytomas and glioblastomas, and correlated inversely with patient overall survival, but the receptor was present in similar levels, independent of tumor grade [218]. The high expression of this chemokine in the most aggressive types of gliomas suggests the involvement of CXC_3_L1-CX_3_CR1 axis in the malignant behavior of these tumors. It was suggested that the upregulation of CX_3_CL1 may be beneficial for neoplastic cells because of adhesive properties of this chemokine when expressed on the cell membrane [218].

### 9.5. CCL2-CCR2 Axis

Pro-angiogenic inflammatory chemokine CCL2, which is normally a monocyte chemoattractant implicated in macrophage recruitment, is also one of important factors in the progression and development of CNS tumors. Physiologically, CCL2 is a regulator of T lymphocytes [222] controlling polarization of Th2 cells into a more immunosuppressive T regulatory phenotype (Treg) in vitro and in animal models [223]. CCL2 also facilitates differentiation of macrophages towards alternatively activated M2-phenotype with pro-tumoral features [224,225].

It was demonstrated that not only normal human brain tissues, but also CNS tumors may express CCL2 constitutively [226]. CCL2 tumor expression was also observed in a rat astrocytoma model [227]. CCL2 is a chemokine secreted by glioma cells to attract microglia and macrophages, which may further promote tumor growth and migration of malignant cells [227]. It was shown that production of CCL2 by glioma cells can be stimulated by ATP [228] and S100B [229]. Moreover, CCL2 expression in vitro was stimulated by pro-inflammatory cytokines IL-1β and TNFα in all glioma cell lines tested [230]. There is a correlation between tumor grade and expression of CCL2. It was revealed that CCL2 is highly expressed in anaplastic astrocytoma and GBM, whereas in fibrillary astrocytoma grade II it is present in lower levels [13,231].

Since CCL2 is believed as the leading chemoattractant for macrophages, it may be suggested that CCL2 produced by glioma may mediate infiltration of these cells into the tumor microenvironment. TAMs could be recruited to tumor site by an interaction between CCL2 (constitutively produced by glioma cells) and its receptor CCR2 (expressed on TAMs) [226,232], what confirms the importance of chemokines as the key players in GBM progression. Significant correlation of CCL2 expression with the degree of macrophage infiltration was shown in human GBM surgical specimens [233]. Similarly, the expression level of this chemokine correlated positively with the TAMs accumulation in tumor mass and its surrounding area [231].

Moreover, TAMs play a pro-angiogenic role in the tumor and secrete CXCL8, contributing additionally to the progression of glioma [231,234]. What is interesting, blockade of CCL2 function with a neutralizing antibody led to a reduction of the infiltration of microglia/macrophages and resulted in prolonged survival in a mice model of gliomas [235]. This indicates that the recruitment of TAMs is strictly related to CCL2 secretion during glioma tumorigenesis. What is important, by releasing MMPs, TAMs may also colonize the pre-metastatic sites, where they promote the seeding and establishment of disseminated tumor cells [236].

Furthermore, it was reported that CCL2/CCR2 axis indirectly promotes tumor progression by increasing the recruitment and suppressive activity not only TAMs, but also MDSCs, the immunosuppressive cells of myeloid origin. MDSCs are known to support the metastatic process by inhibition of antitumor host immunity. CCL2 is also a chemoattractant for MDSCs, which triggers MDSC-dependent suppression of immunological control over the tumor [190,237], stimulates macrophage-mediated angiogenic switch [238,239] and the promotion of tumor growth by these cells [237,240]. It was shown that increased expression of CCR2, a receptor for CCL2, also stimulates the recruitment of TAMs and fibroblasts at the primary tumors, where they enhance invasion, angiogenesis, and metastasis of glioma and other cancers [241] (Figure 5).

### 9.6. CCL5-CCR5 Axis

CCL5, also called RANTES (regulated on activation, normally T-expressed, and secreted) is expressed by T lymphocytes and monocytes/macrophages and is a ligand for chemokine receptor CCR5 [11]. This chemokine stimulates astrocyte proliferation in the early phase of the brain development and regulates the size of astrocyte population [11]. Critical roles of CCL5/CCR5 axis in regulating glioblastoma proliferation and invasion was demonstrated in brain tumors. It was shown that the chemokine CCL5 and its receptor CCR5 play the important roles in GBM development and tumor invasion [242]. High expression of CCR5 was shown in human GBM tissues, where correlated with poor prognosis of patients.

What is more, CCL5-CCR5 axis mediated activation of Akt kinase, inducing the proliferation and invasive responses in glioblastoma cell lines [242], whereas downregulation of CCR5 significantly inhibited tumor growth in a mice model of glioma [242]. Moreover, it was shown that blocking of CCR5 could prevent polarization of macrophages into M2 microglia phenotype, what was associated with a significant reduction in microglia migration, an effect mediated through the inhibition of the AKT pathway [243].

In conclusion, chemokines participate in glioma onset, proliferation, growth enhancement, angiogenesis, its aggressiveness, and influence various pathophysiological mechanisms [1,13]. What is interesting, some chemokines have also the ability to inhibit glioma growth upon specific regulation or interplay with other molecules, including cytokines and growth factors [169] (Table 1).

## 10. Atypical Chemokine Receptors

Apart from conventional chemokine receptors, cCKRs, there is also a distinct group of “atypical” chemokine receptors, which accounts four accepted members, i.e., ACKR1 (previously described as Duffy Antigen for Chemokines or DARC), ACKR2 (or D6 receptor), ACKR3 (or CXCR7), ACKR4 (CCRL1 or ChemoCentryx chemokine receptor, CCX-CKR). There are also two other molecules awaiting functional confirmation, which are tentatively included in the ACKR family. These are ACKR5, also termed as CCRL2, which ligand is non-chemokine chemerin [244], and ACKR6 (or PITPNM3) [245]. ACKRs are mainly expressed at skin and other barrier tissues [104], non-leukocyte cell types, such as erythrocytes and lymphatic or vascular endothelial cells [246]. However, expression of ACKR2 and ACKR3 was also detected on leukocytes [247,248].

ACKRs emerge as a new class of regulators of the highly promiscuous chemokine system and all of them have more than one ligand [249]. Depending on the number of binding ligands, atypical chemokine receptors are categorized as large (ACKR1, ACKR2) or narrow (ACKR3, ACKR4) spectrum receptors. ACKRs were initially termed as “silent” chemokine receptors because of apparent inability to mediate conventional signaling. Activation of these types of receptors largely mediates signaling pathways dependent on β-arrestin. What is important, ACKRs are excluded from classic chemokine signal transduction, although they control the bioavailability of their ligands. This uncoupling of ACKRs from conventional chemokine signaling results in the modulation of ligands bioavailability by internalizing and sequestration of them, chemokine degradation, or their transcytosis in the case of polarized cells [250]. Therefore, ACKRs are also termed as interceptors (internalizing receptors), chemokine-scavenging receptors, or chemokine decoy receptors. Apart from regulation of chemokines, ACKRs may also influence the activity of cCKRs by forming heterodimers or by modulating their expression levels or signaling activity [251].

Atypical receptors for chemokines, together with chemokines and conventional chemokine receptors, might also play a role in the biology of various malignant tumors, including gliomas. Although the role of chemokines and their cCKRS in the pathophysiology of glioma has been extensively studied, less is known about the significance of ACKRs in the development of this type of tumors. However, it was demonstrated that expression of ACKR2 can regulate the micro-environment in some malignant tumors, suggesting a possible role of these receptors in carcinogenesis [252]. Thus, atypical chemokine receptors emerge as a new target for scientific efforts in the search of novel possible biomarkers of malignant tumors. ACKRs may be present not only on cells within the tumor microenvironment, but also directly on cancer cells [104]. ACKRs expression was reported in some leukemias and lymphomas [253,254,255,256], as well as in certain solid tumors, such as breast [257,258,259] and cervical cancer [260], or Kaposi sarcoma [261]. The expression of ACKRs was deregulated in tumor tissues and cancer cells that correlated with their metastatic potential [104].

### 10.1. ACKR1

ACKR1, the oldest known atypical chemokine receptor, controls the chemokine levels and their localization. ACKR1 is a promiscuous chemokine-binding protein, with an ability of joining with ligands from the CC subfamily, including CCL2, CCL5, CCL7, CCL11, CCL13, CCL14, and CCL17. ACKR1 may be also a receptor for multiple chemokines from the CXC subfamily, such as CXCL1, CXCL2, CXCL3, CXCL5, CXCL6, CXCL8, CXCL11 [246]. This receptor was primarily discovered on erythrocytes, as a protein identical with the human blood group antigen, named the Duffy antigen, which may also serve as a co-receptor and cell entrance for the *Plasmodium vivax* [246] and *Plasmodium knowlesi* [262], malarial protozoan parasites. ACKR1 may be also expressed on endothelial cells lining small veins and venules [249]. In addition, presence of this receptor was demonstrated in the cerebellum, exclusively by Purkinje cells [263].

As mentioned above, although ACKRs share seven-transmembrane structural similarity with classical chemokine receptors, they do not participate in signal transduction with G proteins and do not alter intracellular calcium levels. ACKRs do not induce cell migration as well. In fact, ACKR1 lacks the DRYLAIV sequence, which is a highly conserved determinant and plays a critical role in mediating function of GPCRs [12]. Absence of this motif results in ACKR1 inability to couple with G-proteins, therefore ACKR1 fails to transmit detectable intracellular signals and cannot mediate the direct chemokine signaling [250].

It was revealed that ACKR1 may function as a transporter for chemokines across the endothelial cells, what leads to apical retention of ligands and their immobilization [264]. This transcytosis of intact chemokines supports their pro-migratory activity and results in the increase of ligands bioavailability for other CKRs. On the other hand, when expressed on erythrocytes, ACKR1 may reduce chemokines concentration in the circulation, thus acting as a chemokine depot [265]. Although internalization of chemokines by ACKR1 does not lead to their degradation, the receptor may compete with cCKRs for binding of chemokines, thus reducing availability of their ligands. This mechanism of ACKR1 action was proposed to play a role in weakening of chemokine-induced angiogenesis, what may be a protective factor in malignant tumor development [266,267]. ACKR1 may act as a chemokine decoy receptor and interfere with normal tumor growth and chemokine-induced tumor neovascularization. What is more, CXCL8 exerts its pleiotropic effects not only through its conventional receptors CXCR1 and CXCR2, but also through an atypical ACKR1 receptor, which is not coupled with any known ligand-driven signal transduction cascades [170]. It was suggested that by regulation of the excess of chemokines, especially CXCL8, ACKR1 may dampen the pro-angiogenic tumor microenvironment and limit tumor metastasis. Moreover, ACKR1 overexpression in different tumor mouse models of non-small cell lung cancer (NSCLC) led to a decrease in tumor cellularity and vascularity, as well as significantly increased necrosis, what inhibited tumorigenesis and reduced the metastatic capacity of tumor [268].

It was shown that ACKR1 may be related to the development and progression of CNS tumors. Upregulated expression of this receptor was demonstrated in astrocytoma tissues [269]. Importantly, mRNA for chemokine receptors binding CXCL8, both atypical ACKR1 and conventional CXCR1 and CXCR2, were found in all astrocytoma grades tested by reverse transcription/PCR analysis [269]. Furthermore, ACKR1 protein expression was localized on normal brain and tumor microvascular cells, while CXCR1 and CXCR2 were present on tumor infiltrating leukocytes. These findings point out that ACKR1 may contribute to CXCL8-promoted glioma angiogenesis. CXCL8 expression in the early stage of astrocytoma development is initiated by pro-inflammatory stimuli but later in tumor progression it increases due to reduced microenvironmental oxygen pressure. Thus, overexpressed CXCL8 would directly and/or indirectly promote angiogenesis by binding to ACKR1 and through the induction of leukocyte activation and infiltration by binding to its conventional receptors CXCR1 and CXCR2 [269].

### 10.2. ACKR2

ACKR2, or D6 receptor, is more selective than ACKR1 and binds most inflammatory but not homeostatic ligands from the CC subfamily of chemokines, such as CCL2, CCL3, CCL3L1, CCL4, CCL5, CCL7, CCL8, CCL11, CCL12, CCL13, CCL17, and CCL22 [104]. Moreover, in some cases ACKR2 may function as a co-receptor for viral pathogens, such as HIV-1 and HIV-2 [270]. This receptor is expressed constitutively by endothelial cells of lymphatic vessels that drain the skin, the gut, and the lungs [271], on placental trophoblasts [272], and may be also primarily expressed on some subsets of leukocytes, such as innate B-like cells [245], macrophages, dendritic cells, and tissue mast cells [273].

ACKR2 controls chemokine activity and plays a role in their clearance. After binding chemokine, this receptor may also be a transporter of chemokines to intracellular endosomal compartments, where, in contrast to ACKR1, the degradation of ligands occurs [274]. Inside the cell, the ligand dissociates from the receptor. Internalized chemokines remain trapped in the cell and undergo degradation, whereas ACKR2 may be recycled back to the cell surface for further sequestration of its ligands [275]. These repeated rounds of ligand internalization, without reducing ACKR2 levels result in the removing and destroying large amounts of free extracellular proinflammatory chemokines and continuous chemokine sequestration [276,277]. These effects of ACKR2 are also related to its role in the malignant tumor biology and may be related to the reduction of tumor growth and progression. Acting as a CC inflammatory chemokine scavenger, ACKR2 inhibits the recruitment of leukocytes, thus limiting inflammation in vivo. On the contrary, the lack of ACKR2 resulted in accumulation of CC chemokines and infiltration of leukocytes at tumor sites [104]. However, the precise role of ACKR2 in cancer is not fully elucidated yet.

It was shown that cancer cells may express this receptor, both in vivo and in vitro [251]. It was also demonstrated that ACKR2 may play a protective role in breast cancer [257], reducing its proliferation and metastatic potential [278]. Overexpression of ACKR2 inhibited proliferation and invasion of breast cancer cells in vitro as well as reduced lung metastasis in vivo [278]. Furthermore, ACKR2-mediated inhibition of tumorigenesis was associated with decreased levels of chemokines CCL2 and CCL5, lower vessel density, and reduced TAMs infiltration. Additionally, increased ACKR2 expression correlated inversely with tumor severity to lymph node metastasis as well as with clinical stages, but positively correlated to disease-free survival rate in cancer patients [278]. Similarly, by chemokine sequestration, ACKR2 prevents the development of chemically induced malignant skin tumors, while its expression correlated positively with disease-free survival rate [279]. Protective effects of ACKR2 were also demonstrated in Kaposi sarcoma [261]. Nonetheless, this anti-inflammatory activity of ACKR2 not always is beneficial, because this receptor may also lead to a reduced CCR2 expression by NK cells and limiting their infiltration of tumor tissue, what finally supports metastasis [280].

Taken together, these results indicate that the chemokine scavenger activity of ACKR2 may act mainly as protective factor from cancer growth, when recruited leukocytes sustain and enhance the tumor growth. To date, there are no studies that have focused on the importance of ACKR2 in CNS tumors and the role of these receptors in the gliomagenesis is still unexplored.

### 10.3. ACKR3

In contrast to promiscuous ACKR2, ACKR3 is an atypical chemokine receptor with a narrow spectrum of ligands, whose binding is restricted to inflammatory chemokine CXCL11 (which is also a ligand for classic receptor CXCR3) and pro-angiogenic CXCL12 (a ligand for CXCR4) [281,282]. However, a viral chemokine vCCL2/vMIP-II may also act as a ligand for ACKR3 [283]. In addition, ACKR3 may recognize some chemotactic peptides, such as adrenomedullin, intermediate opioid peptide BAM22, and MIF, a pleiotropic cytokine with chemokine-like functions [104].

ACKR3 binds CXCL12 with even higher affinity than the chemokine’s first-described receptor CXCR4 does [284]. Since ACKR3 exhibits large alterations in the DRYLAIV sequence, it cannot activate directly the G-protein-mediated signaling [285]. Like other atypical chemokine receptors, by binding its ligand, ACKR3 induces β-arrestin recruitment [281]. Chemokine scavenging represents the leading mechanism of ACKR3 function. Similarly to ACKR2, ACKR3 also acts as a chemokine scavenger, regulating negatively CXCL11 and CXCL12 [286]. Moreover, it was revealed that CXCL12 binding of ACKR3 may be inhibited by CXCL11 [287]. The functional role of ACKR3 is not only the scavenging its ligands, but also the modulation of CXCR4 expression and function. ACKR3 may regulate CXCL12-mediated G protein signal transmission by forming heterodimers with CXCR4, what can enhance or inhibit CXCR4 expression and function [288].

Constitutive expression of this receptor was detected primarily on venule endothelium and arteriole smooth muscle cells [289], what gives ACKR3 immediate access to circulating CXCL12 and possibility to regulate circulating levels of this chemokine. Hematopoietic cells, lymphatic endothelial cells, mesenchymal cells, and neuronal cells may also express this receptor. ACKR3 expression was also shown on platelets [290,291]. Similarly to other ACKRs, this receptor is also expressed on epithelial tissues of barrier organs [287]. ACKR3 is an essential factor for proper normal embryonic development [292]. Deficiency of this receptor in mice resulted in disturbed cardiac development [293], cardiovascular defects, and early postnatal lethality in those animals [294]. Moreover, a remodeling of cardiac valve is also regulated by receptor ACKR3 functioning [295]. Furthermore, ACKR3 plays a role in the survival of human progenitor neural cells mediated through CXCL12 [20]. By controlling chemokine response, ACKR3 influences also neuronal migration [296], especially facial motor neurons, together with the classical CXCR4 receptor [43], although it seems that their functions in the regulation of interneuron relocation may differ [44].

The ACKR3-CXCL12 axis is involved not only in the developmental and regenerative physiological mechanisms, but also in various pathological processes, including inflammation, regulation of cytokine-driven angiogenesis, tumor cell growth, their survival, adhesion, and invasion in many types of cancers [287]. Constitutive expression of ACKR3 on vascular endothelium cells increases in inflammatory reactions [289]. Likewise, enhanced expression level of ACKR3 was observed on lymphocytes isolated from inflammatory bowel disease patients [297] and on inflammatory phenotype macrophages from atherosclerotic plaques, where it was associated with the synthesis of pro-inflammatory chemokines [298]. In addition, ACKR3 is involved in the enhancement of leukocyte extravasation and promotion of their pro-inflammatory activities, what also promotes immune responses [104]. Taken together, these findings indicate a pro-inflammatory role of this receptor, in contrast to anti-inflammatory activity of AKCR2.

ACKR3 play also a role in various types of cancer. In contrast to other atypical receptors of chemokines, ACKR3 expression by cancer cells promotes the tumorigenesis process in most cases [299]. Upregulation of ACKR3 was found not only on cancer cells in several malignant tumors, such as prostate, breast, and lung cancers, but also on tumor-associated endothelial cells [287,299]. It was also shown that ACKR3 may enhance tumor cell proliferation and inhibit their apoptosis, thus promoting tumorigenesis. In lung cancer, the expression of ACKR3 was the most upregulated among chemokine receptors induced by pro-inflammatory TGF-β1 and correlated with shorter patient survival [300]. Furthermore, TGF-β1 enhanced the migration, invasion, and epithelial–mesenchymal transition of lung adenocarcinoma cells, while ACKR3 knockdown resulted in the reduction of these processes [300]. ACKR3 expression at the protein level was elevated in aggressive prostate tumors, where ACKR3 can increase the expression of pro-angiogenic factors such as CXCL8 and VEGF [301] and support the transendothelial migration of cancer cells [302].

The abovementioned scavenging of CXCL12 by ACKR3 might be expected to have inhibitory effects on the growth and metastasis of cancer cells, opposite to those exerted by the classical CXCL12-CXCR4 axis. Indeed, ACKR3-dependent depletion of CXCL12 and inhibition of short-term CXCR4 signaling was observed in CXCR4-positive breast cancer cells [303]. After ACKR3-mediated internalization, CXCL12 was transported to intracellular compartments, namely lysosomes, where the ligand was degraded, although levels of ACKR3 remained unchanged [303]. However, it was demonstrated that ACKR3, by forming heterodimers with CXCR4 and modifying CXCR4 signaling, may also promote tumor growth and/or metastasis [288,304]. Stimulation of ACKR3 by ligands resulted in the increase of the receptor activity and improved its internalization for transporters and acceleration of its recycling cycles [304]. Additionally, it was revealed that in breast [305] and prostate cancers [306] ACKR3 may form heterodimers of another type, with epidermal growth factor receptor (EGFR), what contributes to promoting tumor cell proliferation in these malignancies.

CXCL11 is a well-recognized ligand for CXCR3, although the interactions of ACKR3 with CXCL11 and its classical receptor, CXCR3, are less known. This chemokine may induce either proliferative or growth-inhibitory signals [75]. Depending on CXCR3 variant, this chemokine may have both pro- and anti-tumorigenic properties. When CXCL11 interacts with ACKR3 or CXCR3-A, it promotes proliferation, whereas binding with CXCR3-B results in growth inhibition [69].

The role of ACKR3 in the development of malignant tumors of CNS, including gliomas was also described by various researchers. It was revealed that in human gliomas, both mRNA and protein levels of ACKR3 were found to be upregulated in glioma [307]. In addition, ACKR3 is highly expressed on tumor endothelial, microglial, and glioma cells [308]. Additionally, a predominant intracellular localization of ACKR3 expression was shown, with only a slight membrane expression in all glioma cell lines examined. Moreover, the receptor was often co-localized with its ligand, CXCL12 [308].

Interestingly, cellular distribution of ACKR3 in gliomas may differ depending on tumor grade. While in grade II gliomas expression of ACKR3 was restricted to tumor cells, in tumor grade III it is present mainly in tumor vascular endothelial cells and only marginally in cancer cells. In glioblastomas (grade IV) enhanced expression of this receptor was found in cancer cells, in the pseudopalisades nearest to necrotic areas and was also exhibited by endothelial cells [309]. In addition, it was revealed that endothelial expression of ACKR3 in human glioma may influence the survival prognosis, depending on the IDH classification. While ACKR3 expression in tumor-associated vessels in IDH1-wildtype tumors predicted a better prognosis, it has reverse consequences in IDH mutated tumors [310]. In addition, a pro-tumoral function of ACKR3 was confirmed in a murine model of glioblastoma ACKR3, where anti-ACKR3 monoclonal antibodies used in combination with temozolomide, a chemotherapy agent, activated immune responses, induced the phagocytic activity of macrophages and the cytotoxic activity of NK cells and complement, what resulted in extended survival of mice [311].

In addition, ACKR3 may be also involved in meningioma tumor cell proliferation and survival. The expression of CXCL11 and CXCL12 as well as both their receptors, atypical ACKR3 and classical CXCR4 were assessed in human meninigioma samples [312]. It was revealed that ACKR3 and CXCL12 were associated with high-proliferative tumors. Moreover, ACKR3 levels correlated to the expression of both ligands examined, what suggests a possible autocrine regulation in this type of CNS tumor. While CXCR4 and CXCL12 were homogeneously expressed within tumor cells, ACKR3 was mainly identified in tumor endothelial cells but CXCL11 in pericytes. These results suggest the possible role of ACKR in meningioma vascularization as well as the preferential expression of ACKR3 and CXCL12 within more aggressive meningioma tumors [312].

### 10.4. ACKR4

ACKR4 is less known in comparison to other receptors in this group. This promiscuous atypical receptor may bind ligands from different chemokine subfamilies: CCL2, CCL8, CCL13, CCL19, CCL21, and CCL25 with high affinity, whereas CXCL13 with low affinity [313]. Interestingly, it seems that mouse ACKR4 cannot bind the latter chemokine, CXCL13 [314]. Similarly to other atypical chemokine receptors, ACKR4 is uncoupled from classic ligand-driven signal transduction cascades, but following chemokine binding, it recruits β-arrestin 2 [315]. ACKR4 is a constitutively internalizing receptor, which acts as a scavenger for the homeostatic chemokines and controls their levels and localization [316]. ACKR4 is expressed on epithelial and lymphatic endothelial cells, thymic epithelial cells, bronchial cells, and keratinocytes [317]. ACKR4 negatively regulates CXCR3-induced chemotaxis by forming complexes with CXCR3 [318]. Moreover, this receptor controls T-cell development in the thymus, suppressing Th17 responses and is responsible for the compartmentalization of CCL21 chemokine in lymph nodes [316].

Like its ligands, ACKR4 is also involved in the progression of malignant tumors. Although ACKR4 is expressed by cancer cells of various malignancies, including breast [319,320], liver [321], and colon cancer [322], it seems that this receptor plays a protective role against tumor development. Both mRNA and protein expressions of ACKR4 correlated with the malignant phenotype of hepatocellular carcinoma (HCC) cells and were significantly reduced in tumor tissue compared with paired normal liver tissue [321]. ACK4 absence was associated with advanced tumor stage and was an independent index for worse survival and increased recurrence [323]. This is in the agreement with the protective effect of ACKR4 in human breast and colon cancer samples, in which ACKR4 downregulation was correlated with worse outcome [324,325]. Moreover, ACKR4 acts as a negative regulator of tumor progression and modulator of metastatic process [324]. By sequestration and targeting its ligands for the degradation, ACKR4 appears to control the bioavailability of these chemokines in vivo. Scavenging of CCL19, CCL21, and CCL25 chemokines, which are the ligands for CCR7 and CCR9, respectively, results in reducing of the migration of HCC cells, which express these conventional receptors [321]. It was also revealed that enhanced expression of ACKR4 inhibited proliferation in vitro of breast cell lines [323]. Similarly, the upregulated overexpression of this receptor in a colon cancer model reduced tumor cell migration and Matrigel invasion [322]. In vivo, beside inhibition of tumor growth, there is also evidence for reduced metastasis breast cancer metastasis and patient survival [323].

Taken together, it seems that ACKR4 expressed by cancer cells acts as a negative regulator of tumor progression and plays an important role as a tumor-suppressive factor. However, the opposite results were also found in some studies, which indicate that ACKR4 could also have a prometastatic role in cancer development [325]. Expression of ACKR4 in mouse mammary cancer cells 4T1.2 resulted in increased spontaneous lung metastasis and enhanced hematogenous metastasis in vivo. Moreover, it was shown in vitro that increased tumorigenicity of ACKR4-expressing 4T1.2 cells were characterized with their more invasive and motile character, lower adherence to ECM and to each other, as well as with the accelerated EMT. Furthermore, analysis of CCX-CKR-expressing 4T1.2 cells also revealed increased expression of TGF-β1, both at mRNA and protein levels. These findings suggest the novel function of ACKR4 as a regulator of expression and the EMT in breast cancer cells. It seems that the role of ACKR4 is more complicated and exceeds beyond its chemokine-scavenging activities. Therefore, further studies are needed to elucidate the importance of ACKR4 in malignant disease. According to our knowledge, there are no data in the literature concerning the importance of ACKR4 in CNS tumors and its function as atypical chemokine receptor in the gliomagenesis is still uncharted.

### 10.5. CCRL2/ACKR5

Chemokine CC-motif receptor-like 2 (CCRL2) is provisionally designated as “ACKR5”, but the confirmation of their chemokine binding specificity and atypical signaling properties is still pending [244,326]. This receptor was also termed as CRAM (chemokine receptor on activated macrophages), although CCRL2 is expressed actually by almost all human hematopoietic cells, including neutrophils, monocytes, macrophages, basophils, mast cells, PMNs, CD4+ and CD8+ T lymphocytes, pro- and pre-B lymphocytes (depending on the maturation stage), dendritic cells, NK cells, and CD34+ progenitor cells, as well as on the cells of barrier tissues, such as epithelium and endothelium [327,328]. CCRL2 molecule has a structure similar to established ACKRs1-4 and is also a 7-domain transmembrane protein lacking DRYLAIV motif, what prevents coupling to the G protein [326]. CCRL2 exhibits a narrow binding spectrum, and it was reported that this receptor may bind homeostatic CCL19 and inflammatory CCL5 chemokines [329,330,331,332] as well as the non-chemokine protein chemerin [333]. Nonetheless, these CCRL2 interactions cannot induce cell chemoattraction or calcium influx because of its apparent inability to signal, similarly to recognized atypical chemokine receptors ACKR1-4 [334]. A constitutive ability of CCRL2 to form homodimers was revealed. Moreover, it was shown that CCRL2 may also regulate functions of other receptors by forming heterodimers with CXCR2, which is a main neutrophil chemotactic receptor. By heterodimerization, CCRL2 influences membrane expression of CXCR2 and promote its functions, including the activation of β2-integrins [335].

Moreover, CCRL2 has the ability to present its ligands [328,333,336,337]. In comparison to other chemotactic signaling receptors ACKR3 and ACKR4, CCRL2 revealed lower capacity to scavenge chemokines, a weaker chemokine internalization, and a slower kinetics of recycling [338]. Similarly to other atypical receptors, CCRL2 may be also coexpressed with its conventional counterparts on leukocytes, interfering with the activation of cCKRs through direct competition for cognate ligands. Coexpression of CCRL2 with CCR7 was observed on B cells and dendritic cells, what resulted in the impairment of CCR7-dependent activation of B cells [253] and dendritic cell chemotaxis [339].

Biological functions of CCRL2 are not fully understood yet. It was initially described to promote chemotaxis in response to CCL2, CCL5, CCL7, and CCL8 chemokines [331,332], but these results are controversial and could not be confirmed by other researchers, since no CCRL2 functional activation by these chemokines was detected. It was suggested that this receptor may participate in immunological inflammatory processes. Studies on CCRL2-knockout mice have revealed that although these animals may develop normally to term and do not show any overt immunological deficits in steady-state conditions, they exhibit some defects in the migration of dendritic cell to peripheral lymph nodes and in mast cells responsible for the reduced Th2 responses and cell-mediated contact hypersensitivity [333,339], what confirms the inflammatory profile of this receptor. Moreover, CCRL2 stimulation by CCL19 did not result in calcium influx or migration, typical effects of chemokine receptor-dependent cellular activation. On the contrary, CCRL2 constitutively recycled and internalized its ligand. Therefore, CCL19-binding competition by CCRL2 might affect the trafficking of lymphocytes and DCs, because CCL19 is critically involved in trafficking these cells in disease states [330].

The CCRL2 receptor is involved not only in inflammatory processes, but also in cancer, although its function in neoplasms is not fully understood at present as well. Depending on the tissue type, it may promote cell migration and invasion [340] or suppress tumor development [259]. Expression of this receptor was observed in several malignant tumors. In breast cancer cells, the constitutive expression of CCRL2 was upregulated in the presence of pro-inflammatory cytokines, such as IL-1β, TNF-α, IL-6, and especially IFN-γ [258]. Moreover, increased levels of CCRL2 were found in tumor tissues with higher immune infiltration. Furthermore, an alternative transcript of *CCRL2* gene, CRAM-A was specifically upregulated after IFN-γ exposure. These results indicate that CCRL2 expression in malignant tissues is specific to inflammatory status [258]. Interestingly, by blocking the activity of CCL2 chemokine, CCRL2 suppressed the migration of breast cancer malignant cells and their invasion [259].

CCRL2 expression was also detected in colorectal cancer, where increased already in the early phase of liver colonization, what suggests involvement of the receptor in tumor metastatic processes [341]. In addition, CCRL2 is overexpressed in a metastatic prostate cancer cells and prostate cancer tissues from patients, both at the mRNA and protein level [342]. Moreover, CCRL2 showed cytoplasmic staining in examined cell line PC-3. Elevated expression of this receptor was also demonstrated on salivary adenoid cystic carcinoma cell lines [343].

According to our knowledge, there is only one study concerning the role of CCLR2 in malignant CNS tumors. It was demonstrated that CCRL2 expression level was significantly elevated in both low grade and high grade human glioma patients and cell lines [340]. Furthermore, in all types tested, i.e., oligodendroglioma WHO grade II, anaplastic oligoastrocytoma WHO grade III, and glioblastoma WHO grade IV, expression level of CCRL2 correlated significantly in a similar way with the expression of its potential ligands, CCL2 and CCL5. CCL2 expression pattern was very similar to that of CCRL2, and significantly increased with higher glioma tumor grade, whereas CCL5 expression exhibited an inverse relationship and its level decreased with increasing tumor grade [340]. Moreover, although CCRL2 did not regulate the growth of human glioblastoma cell lines, its increased expression alone was sufficient to enhance the migration and invasion of glioma tumor cells. These findings suggest that upregulated CCRL2 in glioma promotes cell migration and invasion.

### 10.6. PITPMN3/ACKR6

Phosphatidylinositol transfer protein 3 (PITPMN3) is alternatively named ACKR6, although it is not an accredited atypical chemokine receptor yet. PITPMN3 is a conserved protein of the phosphatidylinositol transfer protein (PITP) family [344]. Proteins from this family contain a phosphatidylinositol transfer protein domain, an acidic region/Ca+2 binding domain. Their structure includes also six transmembrane domains, and a C-terminal domain that interacts with proline-rich tyrosine kinase 2 (Pyk2) [11]. This structure makes PITPMN3 different from other ACKRs, although this receptor is unable to mediate the signal transduction with G proteins [11,345].

PITPMN3 is expressed in premature cells of the retina and in other tissues, such as brain, spleen, and ovaries [346,347,348]. Physiologically, PITPNM3 and its *Drosophila* homologous rdgB are involved in the visual transduction pathway, whereas mutation in human Pyk2-binding domain of PITPNM3 causes autosomal dominant cone dystrophy [349]. It was revealed that PITPNM3 is a functional receptor for CCL18 and activates intracellular calcium signaling, what also differs from other ACKRs. However, this chemokine may also bind to conventional receptor CCR8 and drive leukocyte migration [350].

It was demonstrated that the PITPMN3/CCL18 axis may be implicated in the development of breast, liver, and pancreatic cancer. PITPMN3 may function as an oncogene, when expressed on cancer cells. CCL18 released from TAMs promoted breast cancer metastasis via PITPNM3 [345]. Moreover, it was shown that PITPMN3 may promote invasion of breast cancer cells through the PI3K/Akt/GSK3β/Snail signaling pathway [351]. CCL18 is expressed and released by TAMs in large quantities in human breast cancer [114,352]. In addition, while CCL18 promoted the invasion and metastasis of breast cancer, the suppression of PITPNM3 abolished these effects, what may confirm the critical role of TAM-derived CCL18 in promoting breast cancer metastasis via its atypical chemokine receptor, PITPNM3 [351]. Furthermore, PITPMN3 is also expressed on endothelial cells and upon binding CCL18 may promote angiogenesis and tumor progression. Blocking ACKR6 reduced sprouting of endothelial cells indicating the crucial role of PITPMN3 in the angiogenic effects mediated by CCL18 [114]. In addition, expression of the ACKR6 ligand CCL18 positively correlates with microvessel density/tumor angiogenesis in breast cancer [114]. Expression of PITPMN3 was also shown in hepatocellular carcinoma tissues and correlated with tumor grade and presence of metastases [353]. Furthermore, expression of the PITPMN3 ligand, CCL18, was demonstrated in epithelial cells and TAMs from human pancreatic ductal adenocarcinoma, where correlated with histopathological grading, presence of lymph node metastasis, and overall survival [354]. Briefly, PITPMN3 expression on cancer cells is associated with increased metastasis and worse prognosis. Activated by cytokines released from TAMs, PITPMN3 induces downstream signaling and promotes EMT. According to our best knowledge, there are no records in the literature with any reference to the possible role of PITPMN3/ACKR6 in CNS tumors and its significance in gliomagenesis is still unknown.

In summary, ACKRs emerge as a new group of cancer regulators, which play a role also in CNS tumors’ biology, growth, development, and dissemination. They may be chemokine transporters, presenters, or scavengers, thus regulating chemokine biological activity and availability (Table 2).

## 11. Diagnostic Importance of Chemokines and Their Receptors in Gliomas

Biochemical diagnostics of malignant tumors relies on the use of tumor markers, macromolecular proteinaceous substances, mainly glycoproteins, which are produced by tumor cells or released from normal cells in response to tumor-induced changes. Overexpression of certain genes in the development, proliferation, and differentiation of malignant tumor may result in the increased synthesis of the products of these genes in cancer cells, their release from tumor cells, and elevated concentration levels in body fluids, such as serum/plasma or CSF. The products of these overexpressed genes could serve as the tumor markers useful in the diagnosis and monitoring of CNS tumors.

### 11.1. Serum and Plasma Biomarkers of Gliomas

Elevated serum levels of some classical tumor markers were observed in neuroblastoma patients. One of these biomarkers is carcinoembryonal antigen (CEA), whose concentrations in the sera of patients affected were increased with the progress of the disease [355]. Similarly, some patients with CNS malignant tumors have higher levels of neuron-specific enolase (NSE), an enzyme which is also present in normal cells of the central and peripheral nervous system [356]. As a result of the cell necrosis, often observed within the tumor, NSE is released into the blood and thus may be a marker of neuroendocrine tumors mainly found in children, such as neuroblastoma [356].

However, there are no tumor markers that would be highly sensitive and specific for the most common CNS tumors, such as gliomas. There are also no screening tests used on a regular basis, which could detect tumors in patients that still have no clinical symptoms, or markers useful in differentiation between primary CNS lesions and metastatic brain tumors. Therefore, there new biomarkers of gliomas being pursued, which would be useful especially in the early diagnosis of these CNS tumors, their differentiation, and in predicting the survival of patients. Given the role of chemokines and their receptors in the pathogenesis of glioma, it could be helpful if the detection and measurement of their blood levels were also possible.

Although the expression of various chemokines and/or their receptor were demonstrated in glioma tumor cells or tumor stroma, little is known about serum or plasma levels of these chemokines in glioma patients. There are only a few papers related to chemokine serum concentrations in glioblastoma patients. It was demonstrated that serum levels of CXCL8, the above described chemokine involved in gliomagenesis, were significantly higher in GBM patients, when compared to the healthy subjects [176]. Additionally, one study revealed that serum levels of CXCL12 in patients suffering from malignant glial tumors were significantly elevated in comparison with those in healthy controls [357]. Nonetheless, there were no significant differences between anaplastic astrocytoma and glioblastoma multiforme patients. Similarly, Western blotting did not reveal any differences in the tissue concentration of CXCL12 between patients with these two types of glioma. In another study, the serum levels of CCL2 and CCL5 were also increased in anaplastic astrocytoma and GBM patients, at both mRNA and protein levels [358]. It was demonstrated that serum levels CCL2 and CCL5 were significantly higher in patients suffering from glial tumors than in healthy volunteers, with even five-fold relative increase in CCL5 concentration with regard to controls, what reconfirms the involvement of CCL2 [231,233] and CCL5 [242,243] in the pathogenesis of glial tumors.

### 11.2. Proteomic Profiling in the Search of New Biomarkers

Taking into account that chemokine network is highly complicated, it seems unlikely that any single chemokine or receptor could be a tumor marker sufficiently effective for GBM diagnosis. Therefore, the use of a novel combination of multiple biomarkers was proposed as an alternative attitude to the diagnostics of patients with CNS tumors. This proteome profile should discriminate GBM patients from controls, thus increasing diagnostic accuracy. It was revealed that such proteomic approach may be a useful tool in the selection of potential tumor markers for gliomas, which were subsequently confirmed by sandwich enzyme-linked immunosorbent assay (ELISA) and protein expression in tumoral tissue by means of Western blot [359]. Using SELDI-ToF MS technology, a set of potential serum biomarkers for glioblastoma diagnostic was assessed, finding three proteins, chemokine CXCL4, as well as S100A8 and S100A9 proteins as putative GBM biomarkers, whose serum levels in glioblastoma were increased in comparison with controls [359]. Since each of these proteins characterizes different altered pathways in the development of glioma, combining these markers has advantages over determination of single, individual markers and will provide more accurate information in clinical settings. Therefore, combining them in a multi-marker panel could be more efficient in accurate diagnosis of CNS tumors.

### 11.3. CSF Chemokine Levels as Glioma Tumor Markers

Although the research on traditional sampling sources for proteomic profiling, such as blood and tumor tissue extracts, have generated a considerable volume of evidence, the majority of these potential biomarkers of brain tumors exhibit limited value in clinical situations. The biomarkers for gliomas can be identified in other types of biological samples, not only serum or plasma, but also in CSF, cyst fluid, tumor cell lines or from tumoral tissues. Since the proteomic profiling has emerged as the field of research for the identification of new biomarkers, it became encouraging to look for other types of samples as potential material for determination of tumor markers. CSF appears as a promising sample type for protein biomarker discovery, because it is in close contact with brain tissue and in adjacent vicinity to tumor mass, making it a model reservoir of tumor-related/secreted molecules.

Furthermore, one of the factors that may contribute to the lack of blood biomarkers for CNS malignant tumors is BBB, a barrier which is thought to prevent the release of tumor-specific molecules from brain into the blood circulation. However, the biomarker occurrence in CSF might result not only from the secretion or leaking by tumor tissues, but also from dysfunction of BBB and their increased filtration from peripheral blood [360]. Therefore, CSF may be taken into consideration in the search for more specific brain tumor biomarkers [361], although obtaining CSF for testing has certain limitations from medical and legal points of view [362,363]. On the other hand, extending the diagnostic panel of tests by the possibility to determine chemokines and their receptors as the CSF biomarkers, could be helpful in a better assessment of the clinical condition of patients with brain tumors. As high levels of CCL2 mRNA expression and the presence of CCL2 were observed in malignant glioma cells in studies in vitro [226,230,231], it became an interesting issue whether any changes of this chemokine also appear in CSF from glioma patients. Another question was whether these changes could be useful in a clinical setting, especially in the differentiation of malignant gliomas from benign tumors or even non-tumoral disorders of the CNS, as well as in the detection of subarachnoid dissemination of glioma cells. CCL2 concentrations in CSF samples from patients with malignant gliomas were significantly elevated in comparison with patients with benign or no tumors [362]. Furthermore, significantly higher CCL2 levels were observed in CSF samples from patients with subarachnoid spreading of malignant glioma than in those from patients without dissemination. These findings suggest that the measurement of CCL2 in CSF may lead to more accurate diagnosis, facilitating selection of patients with malignant glioma and dissemination of tumor cells [364].

Importantly, in order to confirm the intrathecal synthesis of any CSF protein concentration, its evaluation should always be performed in relation to the concentrations in serum/plasma, followed by calculation of CSF to serum/plasma indices. Indeed, not only CSF levels of CXCL8, but also their indices were significantly elevated in patients with CNS tumors as compared to the non-tumoral control group [177], what may confirm the potential usefulness of these types of samples in the diagnostic process of glioma management.

## 12. Conclusions

The chemokine network is an extremely complex system, that consists of a large number of interacting ligands and receptors, whose activity often is overlapping. This system regulates diverse cellular processes and is necessary for life, normal functioning, and development. Although the main and first recognized function of chemokine system was chemoattraction of various populations and types of leukocytes, chemokines’ influence extends far beyond this. They play fundamental roles in tissue development, homeostasis, repair of tissues, immunological surveillance and protection from infection, as well as in inflammation and immunity. Furthermore, chemokines and their receptors are involved in various diseases, including malignant tumors. Gliomas are the highly variegated group of malignant tumors of CNS, in which chemokines and their receptors are often upregulated and may play pivotal roles in their development and progression.

## Figures and Tables

**Figure 1 ijms-21-03704-f001:**
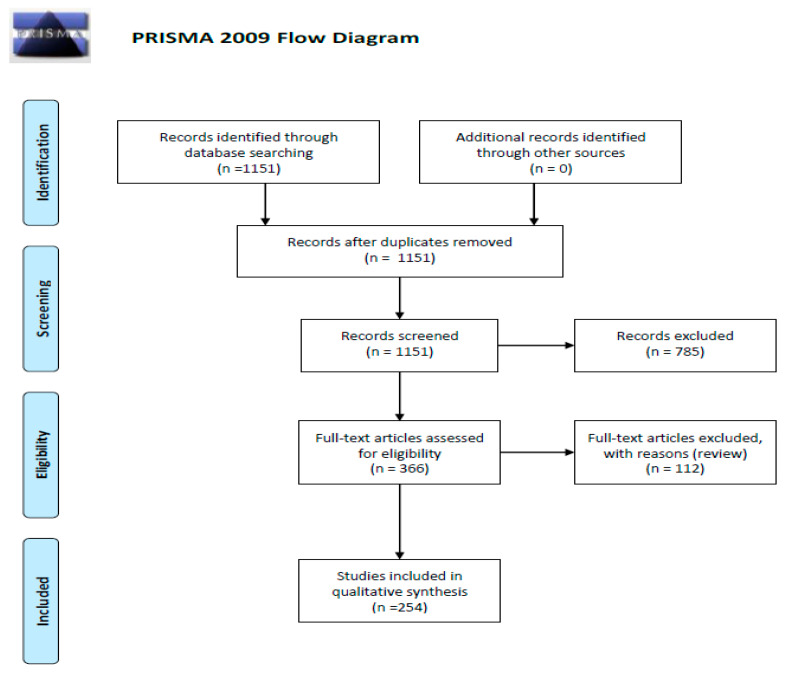
Schematic illustration of articles included in the review manuscript [3].

**Figure 2 ijms-21-03704-f002:**
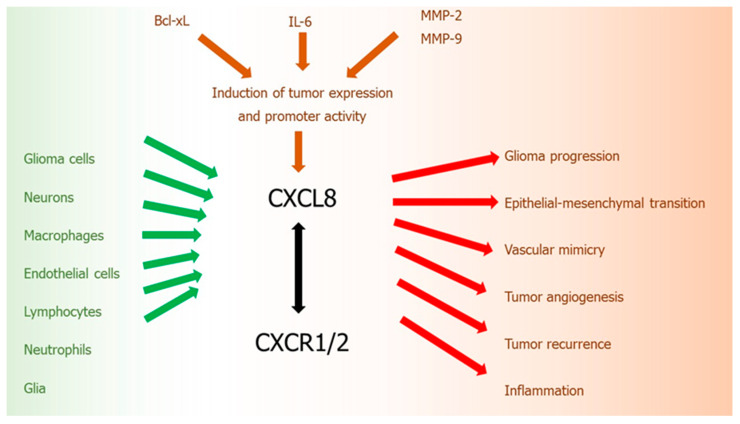
CXCL8-CXCR1/2 axis in glioma development. Green arrows: cells secreting chemokine, red arrows: chemokine activity, brown arrows: factors inducing chemokine, black arrow: chemokine-receptor axis.

**Figure 3 ijms-21-03704-f003:**
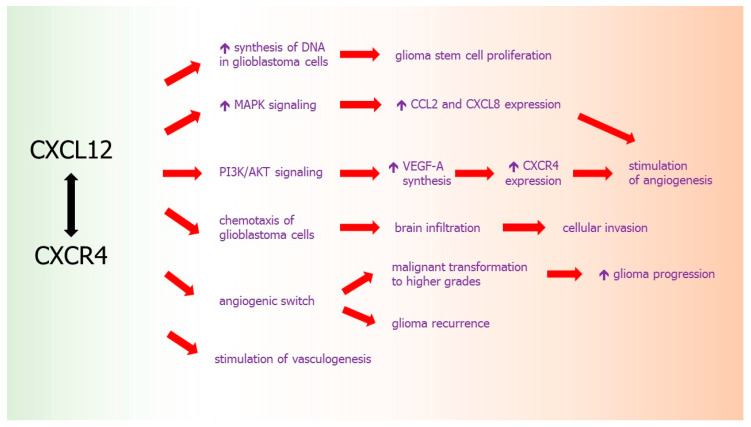
CXCL12-CXCR4 axis in glioma development. Red arrows: effects of chemokine activity, black arrow: chemokine-receptor axis.

**Figure 4 ijms-21-03704-f004:**
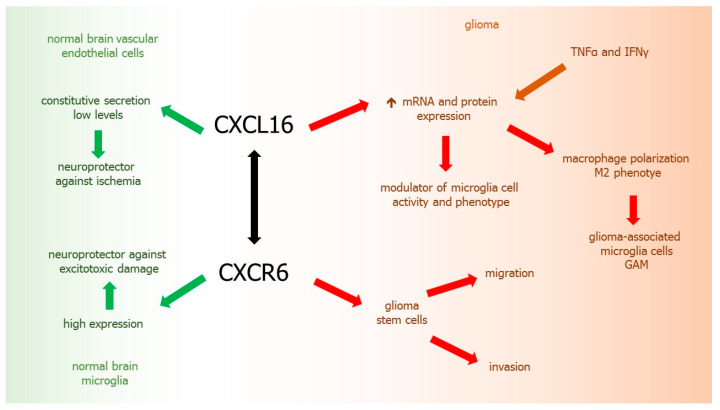
CXCL16-CXCR6 axis in glioma development. Green arrows: chemokine activity in normal brain tissues, red arrows: chemokine activity in glioma, brown arrows: factors inducing chemokine, black arrow: chemokine-receptor axis.

**Figure 5 ijms-21-03704-f005:**
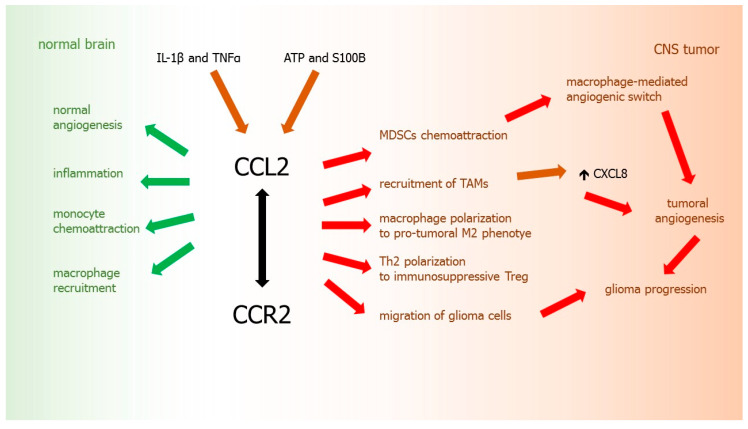
CCL2-CCR2 axis in glioma development. Green arrows: chemokine activity in normal brain tissues, red arrows: chemokine activity in glioma, brown arrows: factors inducing chemokine, black arrow: chemokine-receptor axis.

**Table 1 ijms-21-03704-t001:** Chemokine/receptor axes in gliomas.

Chemokine-Receptor Axis	Importance in CNS Tumors/Gliomas	Author
**CXCL8-CXCR1/2**	CXCL8 expression at mRNA and protein level detected in glioma tissues	[139,170]
	Increased expression of CXCL8 and CXCR2 in tumor samples from GBM patients than in normal brain tissue	[174,188]
	Correlation of CXCL8 and CXCR2 tumor expression with glioma grade and disease-free and overall survival in glioblastoma patients	[174]
	Concentrations of CXCL8 in sera and CSF significantly elevated in patients with CNS tumor compared to non-tumoral control group	[176,177]
	CXCL8-CXCR2 interaction triggers tumor cells proliferation, migration of endothelial cells, and angiogenesis	[139,170,174,187]
	Upregulation of CXCL8-CXCR2 axis in HGG tumors resistant to anti-angiogenic therapy	[183,184]
**CXCL12-CXCR4**	Increased expression of CXCL12 and CXCR4 in various types of malignant CNS tumors	[199,200,201]
	High expression of CXCL12 and CXCR4 localized at the invasive edge and in the regions with high microvessel density, tumor necrosis, and apoptosis within glioblastoma tissue	[201,202,203]
	CXCL12 induces a significant increase of DNA synthesis and chemotaxis in a primary human glioblastoma cell line	[204]
	Correlation of increased CXCR4 expression with tumor grade	[201,207]
	CXCL12 expression as prognostic factor of tumor progression in low-grade gliomas	[209]
	CXCL12-CXCR4 axis participates in tumoral angiogenesis and promotes VEGF production by glioma	[210,211]
**CXCL16-CXCR6**	Increased expression of CXCL16 and CXCR6 at mRNA and protein in human gliomas	[214,215]
	CXCL16-CXCR6 axis promotes human GBM cell growth, invasion, and migration	[215,217]
	CXCL16-CXCR6 axis induces polarization of GAMs toward an anti-inflammatory/pro-tumor phenotype	[51]
**CX_3_CL1-CX_3_CR1**	CX_3_CL1 and CX_3_CR1 expression at mRNA and protein level detected in glioma cells and various grades of glioma	[218,221]
	Correlation of CX_3_CL1 expression levels and glioma grade	[218]
	CX_3_CL1 expression is a prognostic factor of glioma patients’ overall survival	[218]
	Reduced tumor cell aggregation and increased glioma invasiveness as a result of CX_3_CL1 inhibition by monoclonal antibody	[221]
**CCL2-CCR2**	CCL2 secreted by glioma cells promotes tumor growth and migration of malignant cells	[227]
	Correlation of CCL2 expression with tumor grade and TAMs accumulation in GBM tumor mass and its surrounding	[231]
	Blocking of CCL2 with a neutralizing antibody reduced microglia/macrophages infiltrate in glioma and prolonged survival in mice	[235]
	CCL2-CCR2 axis promotes tumor progression by recruitment of suppressive MDSCs	[190,237]
	CCL2 participates in macrophage-mediated angiogenic switch	[238,239]
	Increased expression of CCR2 stimulates TAMs and fibroblasts recruitment at the primary tumors, enhance invasion, angiogenesis, and metastasis of glioma	[241]
**CCL5-CCR5**	CCL5-CCR5 axis induces proliferation and invasive responses in glioblastoma cells	[242]
	High expression of CCR5 in GBM tissues correlates with poor prognosis for patients	[242]
	Downregulation of CCR5 significantly inhibited tumor growth in mice model of glioma	[242]
	Blocking of CCR5 prevents M2 microglia polarization and results in reduced microglia migration	[243]

**Table 2 ijms-21-03704-t002:** Tissue distribution, biological functions, and ligand specificity of atypical receptors for chemokines (ACKRs).

Receptor	Characteristics	
**ACKR1** **(Duffy Antigen for Chemokines, DARC)**	**Ligands**	Pro-inflammatory chemokines: CXCL1, CXCL5, CXCL6, CXCL8, CXCL11, CCL2, CCL5, CCL7, CCL13 [246] Chemokines with mixed function: CCL17 [246] Homeostatic chemokines: CCL14 [246] Non-chemokine ligands: co-receptor for *Plasmodium vivax* [246] and *Plasmodium knowlesi* [262]
	**Tissue/cell expression**	Endothelium of small veins and postcapillary venules [249], erythrocytes [251], Purkinje cells in cerebellum [263]
	**Function**	Chemokine transporter [264] Chemokine depot [265]
	**Importance** **in glioma**	Increased expression in astrocytoma tissues [269] mRNA for ACKR1 detected in various astrocytoma grades [269] Promotion of tumoral angiogenesis by binding CXCL8 [269]
**ACKR2** (D6)	**Ligands**	Pro-inflammatory chemokines: CCL2, CCL3, CCL3L1, CCL4, CCL5, CCL7, CCL8, CCL11, CCL12, CCL13 [104] Chemokines with mixed function: CCL17, CCL22 [104] Homeostatic chemokines: CCL14 [104] Non-chemokine ligands: co-receptor for HIV-1 and HIV-2 [270]
	**Tissue/cell expression**	Lymphatic endothelium of gut, lung, and skin [271], placental trophoblast [272], hematopoietic stem cells [249], leukocytes: innate B cells [245], macrophages, dendritic cells, tissue mast cells [273]
	**Function**	Chemokine scavenger [275] Anti-inflammatory activity [276,277]
	**Importance** **in glioma**	Not described yet
**ACKR3** (CXCR7)	**Ligands**	Homeostatic chemokines: CXCL11 [281], CXCL12 [282] Non-chemokine ligands: Adrenomedullin, MIF, BAM22 [104] Viral chemokine: vCCL2/vMIP-II [283]
	**Tissue/cell expression**	Smooth muscle cells of venules and arterioles [289], leukocytes: T lymphocytes, B lymphocytes, dendritic cells [289], epithelium [287]
	**Function**	Chemokine scavenger [286] Modulator of CXCR4 receptor [288]
	**Importance** **in glioma**	High expression in glioma tumor cells [307], microglia, and tumor-associated vascular endothelium [308] Expression in glioma related to the survival prognosis [310] Autocrine regulator of meningioma development and vascularization [312]
**ACKR4** (CCRL1/CCXCKR)	**Ligands**	Pro-inflammatory chemokines: CCL2, CCL8, CCL13, CCL19 [314] Chemokines with mixed function: CCL21 [314,316] Homeostatic chemokines: CCL25 [313,314], CXCL13 [313]
	**Tissue/cell** **expression**	Leukocytes, dendritic cells, T cells [313], lymph nodes, spleen, thymus [317] Non-lymphoid tissues: heart, kidney, placenta, trachea, and brain [313] Epithelium, bronchial cells, keratinocytes [317]
	**Function**	Chemokine scavenger [315,316,317] Modulator of CXCR3 receptor by forming complexes [318]
	**Importance** **in glioma**	Not described yet
**ACKR5 *** (CCRL2)	**Ligands**	Pro-inflammatory: CCL19 [330], CCL2, CCL5, CCL7, and CCL8 [329,331,332] Non-chemokine ligands: Chemerin [333]
	**Tissue/cell** **expression**	Leukocytes: neutrophils, monocytes, macrophages, basophils, mast cells, PMNs, CD4+ T cells, CD8+ T cells, pro- and pre-B cells, dendritic cells, NK cells, CD34+progenitor cells, epithelium, endothelium [327,328]
	**Function**	Chemokine presenter [328,333,336,337]. Regulation of other receptor functions by forming heterodimers [335].
	**Importance** **in glioma**	Elevated expression in human glioma samples and cell lines [340] Upregulated CCRL2 in glioma promotes cell migration and invasion [340]
**ACKR6 *** (PITPNM3)	**Ligands**	Pro-inflammatory: CCL18 [346]
	**Tissue/cell** **expression**	Retina, brain, spleen, ovaries [346,347,348], tumor-associated macrophages [114,352]
	**Function**	Activator of intracellular calcium signaling [348,350]
	**Importance** **in glioma**	Not described yet

*-not confirmed as atypical receptor yet, tentatively included in the ACKR family.

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
