# Peer review of "The Role of Selected Chemokines and Their Receptors in the Development of Gliomas"

_ijms, 2020, doi:10.3390/ijms21103704_

Round 1
Reviewer 1 Report
In this review, Magdalena Groblewska, Ala Litman-Zawadzka and Barbara Mroczko described the role of chemokines and their receptors in the development of gliomas. This is an interesting review, with a lot of descriptions.
I would accept this manuscript if the authors add some figures in which they represent the receptors and the corresponding chemokines, and their role in glioma development. This will help the readers to better understand the complex role of this family of receptors.
Author Response
Response to the Reviewer 1:
In this review, Magdalena Groblewska, Ala Litman-Zawadzka and Barbara Mroczko described the role of chemokines and their receptors in the development of gliomas. This is an interesting review, with a lot of descriptions.
Thank you very much for this general positive comment.
I would accept this manuscript if the authors add some figures in which they represent the receptors and the corresponding chemokines, and their role in glioma development. This will help the readers to better understand the complex role of this family of receptors.
Thank you very much for this remark. These Figures have been inserted in appropriate places in the new version of the manuscript (pages 14, 18, 19, and 21).
Reviewer 2 Report
This review paper described the cytokines and their receptors in the gliomas.
However, it is not acceptable in this form.
- Abstract: it is too general. It is better to describe the specific properties in the gliomas.
- Text: It is too long to describe the contents about general cytokines & their receptors. It is better to be concise in focus of them in gliomas.
- Tables: It is not clear to see and also please add reference to them.
- It will be better to depict figures about cytokines & receptors in gliomas.
Author Response
Response to the Reviewer 2:
This review paper described the cytokines and their receptors in the gliomas.
However, it is not acceptable in this form.
- Abstract: it is too general. It is better to describe the specific properties in the gliomas.
Thank you for this comment, the abstract has been changed according to the Reviewer’s suggestions.
- Text: It is too long to describe the contents about general cytokines & their receptors. It is better to be concise in focus of them in gliomas.
Thank you very much for this valuable suggestion. We have revised and shortened this part of the manuscript. All changes have been highlighted in whole manuscript.
- Tables: It is not clear to see and also please add reference to them.
Thank you very much for this remark. These Tables have been rearranged and references have been added in the new version of the manuscript (pages 22-23 and 32-33).
- It will be better to depict figures about cytokines & receptors in gliomas.
Many thanks for your valuable suggestions. These figures have been added in the revised version of the paper (pages 14, 18, 19, and 21).
Round 2
Reviewer 1 Report
Thank you for adding the figures but legends would help to better understand the arrow colors, the background colors.
Figure 3 is really hard to read, please simplify it.
A quick revision of all figures would be appreciated. Same typo size, same font...
Author Response
Response to the Reviewer 1:
Thank you for adding the figures but legends would help to better understand the arrow colors, the background colors.
Thank you very much for this remark. Figure legends have been added in the new version of the manuscript (pages 13, 18, 19, and 21). The background colors are not attributed to anything special feature od chemokines or cells, but arrow colors are explained in the revised version of the manuscript.
Figure 3 is really hard to read, please simplify it.
Thank you for this comment. We have prepared simplified version of this figure.
A quick revision of all figures would be appreciated. Same typo size, same font...
Thank you very much for your valuable suggestion. All figures have been revised.
Reviewer 2 Report
This revised form is now acceptable for publication of this journal, IJMS.
Author Response
Response to the Reviewer 2:
This revised form is now acceptable for publication of this journal, IJMS.
Thank you very much for the acceptance of our manuscript.